# Contact-separation-induced self-recoverable mechanoluminescence of CaF$_2$:Tb$^{3+}$/PDMS elastomer

Wenxiang Wang[1], Shanwen Wang[1], Yan Gu[1], Jinyu Zhou[1] & Jiachi Zhang [1]✉

Centrosymmetric-oxide/polydimethylsiloxane elastomers emit ultra-strong non-pre-irradiation mechanoluminescence under stress and are considered one of the most ideal mechanoluminescence materials. However, previous centrosymmetric-oxide/polydimethylsiloxane elastomers show severe mechanoluminescence degradation under stretching, which limits their use in applications. Here we show an elastomer based on centrosymmetric fluoride CaF$_2$:Tb$^{3+}$ and polydimethylsiloxane, with mechanoluminescence that can self-recover after each stretching. Experimentation indicates that the self-recoverable mechanoluminescence of the CaF$_2$:Tb$^{3+}$/polydimethylsiloxane elastomer occurs essentially due to contact electrification arising from contact-separation interactions between the centrosymmetric phosphors and the polydimethylsiloxane. Accordingly, a contact-separation cycle model of the phosphor–polydimethylsiloxane couple is established, and first-principles calculations are performed to model state energies in the contact-separation cycle. The results reveal that the fluoride–polydimethylsiloxane couple helps to induce contact electrification and maintain the contact-separation cycle at the interface, resulting in the self-recoverable mechanoluminescence of the CaF$_2$:Tb$^{3+}$/polydimethylsiloxane elastomer. Therefore, it would be a good strategy to develop self-recoverable mechanoluminescence elastomers based on centrosymmetric fluoride phosphors and polydimethylsiloxane.

Mechanoluminescence (ML) is a mechanic-photon conversion process in which materials emit light under external mechanical stimuli, such as friction, stretching, compression, and impact[1–3]. Compared with other types of luminescence, ML can transduce the ubiquitous mechanical energy that occurs in our daily lives to generate light, avoiding the requirement of artificial excitation sources[4–6]. Due to its unique mechanic-photon conversion nature, ML has attracted widespread attention and is widely investigated for applications in sensors, anticounterfeiting, displays, imaging, lighting and intelligent wearable devices[7–9].

In the past few decades, many ML materials have been reported[10,11]. The most well-known ML materials are asymmetric piezoelectric SrAl$_2$O$_4$:Eu$^{2+}$ and ZnS:Cu/Mn$^{2+}$[12–16]. However, SrAl$_2$O$_4$:Eu$^{2+}$ requires preirradiation to charge traps before emitting ML[17–20]. ML of ZnS:Cu/Mn$^{2+}$ does not require preirradiation. Particularly, Jeong and Choi reported the first demonstration of 100,000 MLs in ZnS:Cu/polydimethylsiloxane (PDMS) films without preirradiation[21]. In 2013, Chandra first reported that the "self-recovery" ML of the sulfides takes place by trapping of drifting carriers in a piezoelectric field[22]. Consequently, ML displays seem to be becoming a reality, and piezoelectric sulfides are considered the best ML materials. However, the sulfides are unstable and lack multicolor luminescence[23–25]. Recently, our group reported a series of ultra-strong multicolor ML elastomers based on PDMS polymers and centrosymmetric oxide phosphors such

---

[1]National & Local Joint Engineering Laboratory for Optical Conversion Materials and Technology, Lanzhou University, Lanzhou, P. R. China.
✉e-mail: zhangjch@lzu.edu.cn

**Fig. 1 | Structural characteristics of the CaF$_2$:Tb$^{3+}$ phosphor and CaF$_2$:Tb$^{3+}$/ PDMS elastomer. a** Schematic diagram and scanning electron microscopy (SEM) (inset) of the elastomer; **b** microscope and element distributions of the CaF$_2$:Tb$^{3+}$ particles in the elastomer (the scale bars is 100 μm); **c** Rietveld structure refinements of the CaF$_2$:Tb$^{3+}$ phosphor (the "a. u." is the abbreviation of "arbitrary units"; source data are provided as a Source Data file); **d** HAADF-STEM image of the CaF$_2$ crystal (the scale bars are 2 nm in the figure and 500 pm in the inset; the red spheres in the inset represent the Ca atoms); **e** schematic diagram of a piezoelectric structure (I) and a centrosymmetric non-piezoelectric structure (II) under stress

(the red/blue circles represent cations/anions and the bi-colored middle circles represent positive/negative charge centers); **f** phase (I) and amplitude (II) of piezoresponse force microscopies for the CaF$_2$:Tb$^{3+}$ phosphor (the red and black data points represent the signal intensity when the tip voltage increases and decreases, respectively; the "a. u." is the abbreviation of "arbitrary units"; source data are provided as a Source Data file). HAADF-STEM, high-angle angular dark field-scanning transmission electron microscopy; PDMS polydimethylsiloxane; SEM scanning electron microscopy.

as Sr$_3$Al$_2$O$_6$:Eu$^{3+}$, Y$_3$Al$_5$O$_{12}$:Ce$^{3+}$, Lu$_3$Al$_5$O$_{12}$:Ce$^{3+}$, CaZnGe$_2$O$_6$:Mn$^{2+}$, ZrO$_2$:Ti$^{3+}$, and Gd$_5$Ga$_3$O$_{12}$:RE$^{3+}$ [26–31] (Please refer to Supplementary Note 1 for the reason why we focused on centrosymmetric phosphors). Significantly, these centrosymmetric oxide phosphors are sufficiently stable, and the ML of these phosphor/PDMS elastomers does not require any preirradiation. Moreover, the PDMS-based elastomer is flexible, stretchable, and generally considered one of the most ideal ML materials for intelligent wearable devices [32–34]. Unfortunately, there is still a fatal drawback for these excellent centrosymmetric-oxide/ PDMS elastomers—ML degradation occurs under stretching [35]. Although the ML brightness of these elastomers can self-recover and remains stable after each slight scratching, it fails to recover after each hard stretching. Particularly, after being hard stretched only three times, the ML brightness of these elastomers will significantly decrease to almost zero and cannot recover in a short time due to interface damage. Briefly, the term "damage" refers to the tiny gaps generated at the phosphor–PDMS interface, generally created by the separation of the centrosymmetric phosphors and PDMS during the contact–separation cycle. These tiny gaps at the interface can hinder the next contact between the phosphors and the PDMS, thus affecting the ML self-recoverability of the elastomer. Previously, we attempted many methods to improve these centrosymmetric-oxide/PDMS elastomers, but none were able to achieve self-recoverable ML under stretching. Since ML elastomers inevitably need to be stretched, this fatal drawback of prior elastomers greatly limits their use in practical applications [31]. Therefore, it is necessary to develop a new type of flexible centrosymmetric phosphor/PDMS elastomer with stable self-recoverable ML under stretching.

In this work, we present a flexible ML elastomer based on a centrosymmetric fluoride phosphor CaF$_2$:Tb$^{3+}$ and PDMS polymers. The CaF$_2$:Tb$^{3+}$/PDMS elastomer exhibits partially self-recoverable ML after each stretching. This is the first time we have observed that ML

can self-recover to approximately 60–75% of the ML intensity at previous stretching. Significantly, the bright green ML of the CaF$_2$:Tb$^{3+}$/PDMS elastomer can be clearly observed even after being stretched more than 30 times. Therefore, although the material still needs further improvement, it is very valuable to study in detail the unique self-recoverable ML mechanism of the CaF$_2$:Tb$^{3+}$/PDMS elastomer under stretching. Investigations indicate that the ML of the CaF$_2$:Tb$^{3+}$/PDMS elastomer originates from contact electrification arising from contact–separation interactions between the centrosymmetric phosphorus and the PDMS under mechanics. Based on contact electrification, charges are transferred during contact and produced during separation [30]. Accordingly, a contact–separation cycle model at the interface of centrosymmetric-phosphor/PDMS is established, and the energies of each state in the contact–separation cycle are calculated. This procedure demonstrates that the energy at the fluoride/PDMS interface is low enough that the fluoride phosphors can restore close contact after separation from the PDMS, resulting in significantly less damage at the interface and self-recoverable ML under stretching. The significant result can guide the improvement of self-recoverable ML for centrosymmetric phosphor/ PDMS elastomers.

## Results

Figure 1 presents the structural characteristics of the CaF$_2$:Tb$^{3+}$ phosphor and CaF$_2$:Tb$^{3+}$/PDMS elastomer (Supplementary Note 2). Figure 1a shows that the elastomer is fabricated based on the CaF$_2$:Tb$^{3+}$ phosphor and PDMS polymers, and the thickness of the elastomer is approximately 500 μm, as shown in the inset. As shown in Fig. 1b, the phosphor particles are uniformly distributed in the PDMS matrix (I), and the Ca, F, and Tb elements are uniformly dispersed in phosphor particles (II–IV) [36]. According to the density functional theory (DFT) calculated cohesive energies ($E_c$) for the different defects in the CaF$_2$

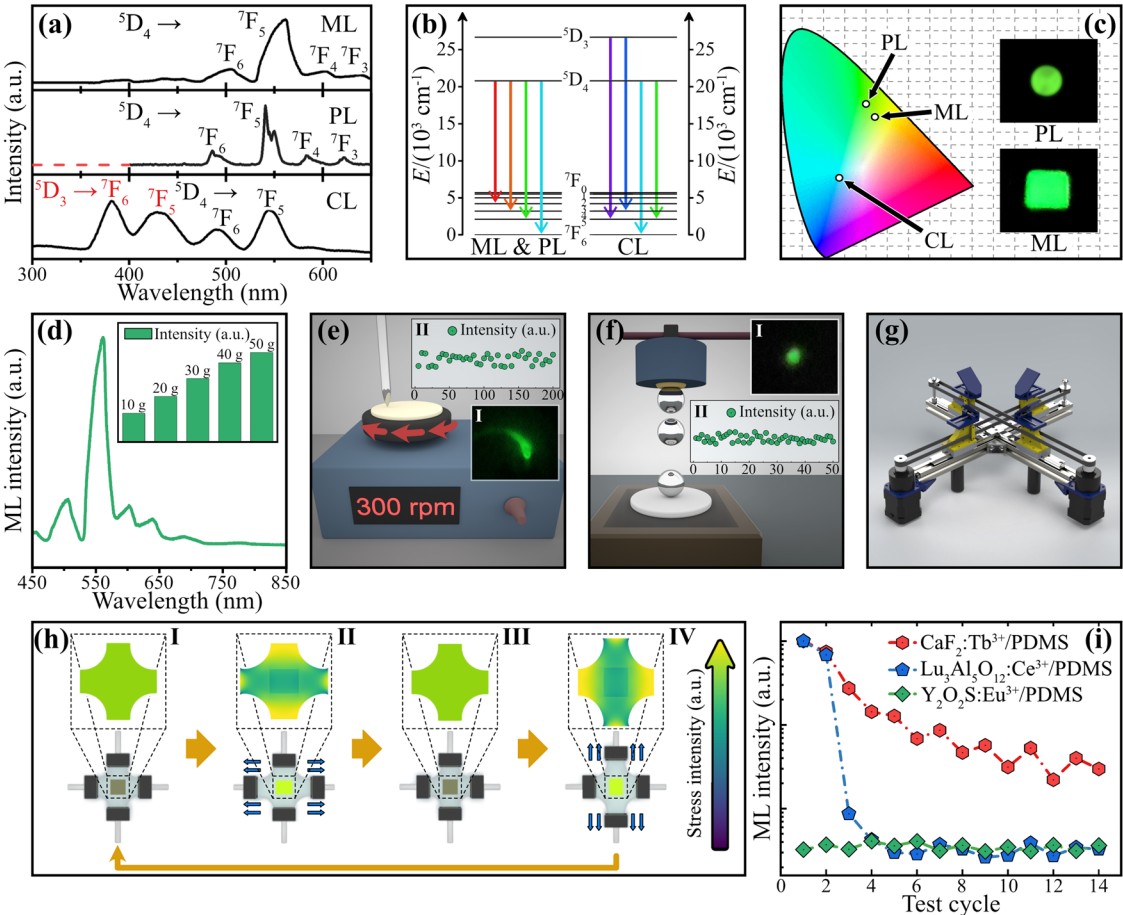

**Fig. 2 | ML properties of the CaF₂:Tb³⁺/ PDMS elastomer. a** ML, PL, and cathode-ray luminescence (CL) spectra of the elastomer (the $^5D_{3-4}$ and $^7F_{0-6}$ represent the excited and ground levels of Tb³⁺, respectively; the "a. u." is the abbreviation of "arbitrary units"; source data are provided as a Source Data file); **b** ML, PL, and CL emission transitions of Tb³⁺ emitters (the colored arrows represent the emissions due to the $^5D_{3-4} \to {}^7F_{0-6}$ transitions); **c** chromaticity coordinate (CIE 1931) of the ML, CL, and PL spectra; **d** ML spectrum under scratching and ML intensities (inset) of the elastomer under different stresses (the stresses corresponding to load masses on the nail; the "a. u." is the abbreviation of "arbitrary units"; source data are provided as a Source Data file); **e** a friction machine to test the ML of the elastomer and ML photo (I)/intensities (II) under scratching (the nail is fixed above and the platform rotates in the direction of the arrow at a speed of 300 rpm; the "a. u." is the abbreviation of "arbitrary units"; source data are provided as a Source Data file); **f** a falling ball experimental device to test the ML of the elastomer and ML photo (I) /

intensities (II) under hitting (a 15.0 g iron ball is released by an electromagnet and falls freely from a height of 0.5 m; the "a. u." is the abbreviation of "arbitrary units"; source data are provided as a Source Data file); **g** a two-dimensional stretching machine to test the ML of the elastomer under stretching; **h** schematic diagram and finite element stress (von Mises) simulations of the elastomer under two-dimensional stretching: the elastomer (brown block) is held in place by four shelves (black block) on a two-dimensional stretching machine and stretched along two orthogonal axes (gray); the blue arrow represents the stretching direction, and the yellow arrow represents the cycling direction; the color scale arrow indicates the stress intensity, which increases in the direction of the arrow; **i** normalized ML intensities of the CaF₂:Tb³⁺/PDMS, Lu₃Al₅O₁₂:Ce³⁺/PDMS, and Y₂O₂S:Eu³⁺/PDMS elastomers under continuous stretching (the "a. u." is the abbreviation of "arbitrary units"; source data are provided as a Source Data file). ML mechanoluminescence; PDMS polydimethylsiloxane.

crystal (Supplementary Note 3), two Tb³⁺ dopants would replace three Ca²⁺ cations, resulting in two [Tb$_{Ca}$'] and one Ca²⁺ vacancy [V$_{Ca}$"] to keep the charge balance in the CaF₂ crystal. According to the Rietveld structure refinements in Fig. 1c (Supplementary Tables 4, 5), the CaF₂ crystal belongs to a cubic centrosymmetric structure with a space group of Fm-3m[37,38]. Moreover, Fig. 1d shows the high-angle angular dark field-scanning transmission electron microscopy (HAADF-STEM) image of the (1 1 1) crystal plane of CaF₂. Periodically appearing bright spots are observed along the direction of the arrow corresponding to calcium atoms (inset i), further demonstrating the long-range ordered centrosymmetric structure of the CaF₂ crystal[39–41]. Accordingly, the positive and negative charge centers of the centrosymmetric CaF₂ crystal always coincide under stress, as shown in Fig. 1e-II, such that it cannot generate charge due to the piezoelectric effect (Fig. 1e-I)[42–44]. As experimental evidence, a typical overlapping hysteresis loop is observed in piezo-response force microscopies of the CaF₂:Tb³⁺ phosphor (Fig. 1f), which further demonstrates that the CaF₂:Tb³⁺

phosphor is nonpiezoelectric and that Tb³⁺ doping does not significantly break the symmetry of the CaF₂ crystal[45–47].

Figure 2 presents the ML properties of the CaF₂:Tb³⁺/PDMS elastomer. Significantly, the CaF₂:Tb³⁺/PDMS elastomer can emit intense green ML without being preirradiated, which is similar to the well-known performance of the piezoelectric ZnS/Mn²⁺/Cu[48,49]. Notably, Fig. 2a shows that the ML spectrum of the elastomer is identified as the $^5D_4$–$^7F_j$ transitions of Tb³⁺, which is similar to its photoluminescence (PL) spectrum (Supplementary Note 4)[50–52]. This result indicates that the ML and PL follow similar emission paths (Fig. 2b), resulting in similar emission colors (Fig. 2c). However, the ML spectrum is clearly different from the cathode-ray luminescence (CL) spectrum composed of intense $^5D_3$–$^7F_j$ transitions of Tb³⁺, which demonstrates that the ML is not due to electron bombardment, which is typically responsible for the CL[53]. In addition, the width of the ML, PL, and CL spectra varies significantly due to differences in the spectrophotometer and the measurement slit. Figure 2d depicts the ML spectra and intensities

(inset) of the $CaF_2$:$Tb^{3+}$/PDMS elastomer under different stresses. Apparently, in a certain range, the ML intensity of the elastomer linearly increases with increasing applied stress, showing potential application for stress-photon sensing[54–56]. To investigate the ML self-recoverability of the $CaF_2$:$Tb^{3+}$/PDMS elastomer under continuous scratching, a friction machine controlled by a motor is applied, as shown in Fig. 2e, and the stress load on the nail is set to 30 g. The green ML (I) of the elastomer under continuous scratching is very stable, and the ML intensity can completely self-recover even after being scratched 200 times (II), demonstrating favorable ML self-recoverability of the $CaF_2$:$Tb^{3+}$/PDMS elastomer under continuous scratching. In addition, a falling ball experiment is also conducted, as shown in Fig. 2f. A 15.0 g iron ball is transported to the top of the elastomer with an electromagnet. Then, the iron ball falls freely from a height of 0.5 m, then strikes the elastomer and excites bright green ML, as shown in Fig. 2f (I). Figure 2f (II) indicates that the ML of the $CaF_2$:$Tb^{3+}$/PDMS elastomer can self-recover well under continuous hitting, and it can keep stable after being hit 50 times, demonstrating the excellent self-recoverability of this elastomer under continuous hitting.

To investigate the ML self-recoverability of the $CaF_2$:$Tb^{3+}$/PDMS elastomer under continuous stretching, we manufacture a two-dimensional stretching machine controlled by a microcontroller with a controllable stretching accuracy of 0.1 mm (Supplementary Note 5), as shown in Fig. 2g. Accordingly, Fig. 2g depicts the schematic diagrams and finite element stress (von Mises) simulations of the $CaF_2$:$Tb^{3+}$/PDMS elastomer under two-dimensional stretching. To avoid measurement error caused by uneven stress on the edge, the elastomer is fabricated into a special four-leaf clover shape. Consequently, the edge region where the elastomer is clamped is pure PDMS (Supplementary Note 6), and the $CaF_2$:$Tb^{3+}$ phosphor is mainly distributed in the rectangular region at the center of the elastomer, as shown in Fig. 2h. Based on the results of the finite element simulation, the stress distributed on the central rectangle of the elastomer is uniform and symmetrical in both transverse and longitudinal stretching. To quantitatively analyze the ML properties of the elastomers under stretching, the stretching speed, distance, and frequency are set to 10.0 mm/s, 6.0 mm, and 1.0 Hz. Accordingly, Fig. 2i presents a comparison of the normalized ML intensities of the $CaF_2$:$Tb^{3+}$/PDMS elastomer and typical elastomers based on other centrosymmetric phosphors, such as $Lu_3Al_5O_{12}$:$Ce^{3+}$ and $Y_2O_2S$:$Eu^{3+}$ (Supplementary Note 7). To rule out the potential influence of fluorescent or sunlight exposure, all experiments in this work were carried out in a dark room illuminated by red light-emitting diode (LED) lamps. Moreover, the results demonstrate that the ML properties of the $CaF_2$:$Tb^{3+}$/PDMS elastomers are completely unaffected even when exposed to ultraviolet or blue light (please refer to Supplementary Note 8 for more details). The results in Fig. 2i indicate that the ML of the $Y_2O_2S$:$Eu^{3+}$/PDMS elastomer is negligible, even under intense stretching, and the ML of the $Lu_3Al_5O_{12}$:$Ce^{3+}$/PDMS elastomer is sharply reduced to almost zero after stretching only three times. While the ML of the $CaF_2$:$Tb^{3+}$/PDMS elastomer also decreases after each stretching, it can still self-recover to approximately 60–75% of its ML intensity at previous stretching, which is significantly more stable than the competing materials. Consequently, the bright green ML of the $CaF_2$:$Tb^{3+}$/PDMS elastomer can be clearly observed even after being stretched more than 30 times. According to the above results, the $CaF_2$:$Tb^{3+}$/PDMS elastomer shows a completely self-recoverable ML under slight scratching and a partially self-recoverable ML under hard stretching. Generally, the entire elastomer is stressed under stretching, while only a very tiny area in contact with the nail is stressed under scratching. Even if the same pressure is applied, the stress under stretching must be much greater than the stress under scratching due to the much larger stress-bearing area. Therefore, we always feel that the stress under scratching is "slight" and the stress under stretching is "hard", even under the same pressure (Supplementary Note 9).

To understand the self-recoverable ML of the $CaF_2$:$Tb^{3+}$/PDMS elastomer, it is necessary to first understand its ML mechanism. As mentioned above, the $CaF_2$:$Tb^{3+}$ belongs to a typical nonpiezoelectric centrosymmetric structure, and its PDMS-based elastomer does not require preirradiation before emitting ML. Therefore, the ML of the $CaF_2$:$Tb^{3+}$/PDMS elastomer should not be due to the traditionally believed piezoelectric effect and detrapping of traps[57,58]. Moreover, it is significant to note that the $CaF_2$:$Tb^{3+}$ phosphor powders never emit any ML under mechanics unless they are compounded into the PDMS. This dynamic suggests that the friction interactions at the interface between the centrosymmetric phosphor and the PDMS should be responsible for ML generation (Supplementary Note 10). Figure 3a depicts the change in ML intensity and stretching distance of the $CaF_2$:$Tb^{3+}$/PDMS elastomer during a single stretching. This result indicates that ML does not occur from the moment the elastomer begins to be stretched. ML occurs only when the elastomer has been stretched for approximately 65% of its maximum stretching distance. Moreover, the ML intensity reaches its maximum within 0.1 s and rapidly decreases to almost zero before the elastomer reaches its maximum stretching distance. This result demonstrates that the ML of the $CaF_2$:$Tb^{3+}$/PDMS elastomer only occurs once during the stretching process, more possibly at the moment of the separation of the phosphor–PDMS interface, which is clearly different from the piezoelectricity-induced ML (the ML performance of the typical ZnS:Cu/PDMS elastomer during a single stretching is presented in Supplementary Note 11). To demonstrate the separation of the interface, the $CaF_2$:$Tb^{3+}$/PDMS elastomers before stretching (I) and after stretching for 20% (II) and 100% (III) of the maximum stretching distance are immersed in red eosin Y alcohol solution (1%). The microscopes in Fig. 3b indicate that only the $CaF_2$:$Tb^{3+}$/PDMS elastomer that has been stretched 100% of the maximum stretching distance is significantly dyed red. It is well known that the separation of two insulators in close contact will produce charges due to the contact electrification effect[59–61]. Accordingly, we propose that the contact electrification arising at the interface between the insulated centrosymmetric phosphor and the PDMS polymer is involved in the ML of the elastomer. To demonstrate the charge distribution, Fig. 3c presents the surface morphology and the surface electron distribution of the $CaF_2$:$Tb^{3+}$/PDMS elastomer. The bright spots in Fig. 3c (I) correspond to the $CaF_2$:$Tb^{3+}$ phosphor particles on the elastomer. The electrons sprayed from the left side are mainly concentrated in the phosphor particles, as shown in Fig. 3c (II). This result demonstrates that the fluoride phosphor attracts electrons more easily than the PDMS polymer due to its stronger electronegativity. Therefore, the contact-electrification-induced electrons are transferred to the fluoride phosphor when the $CaF_2$–PDMS interface is separated[62–64]. According to the triboelectric series, PDMS is relatively electronegative compared to the most common polymers. However, the inorganic $CaF_2$:$Tb^{3+}$ phosphors containing highly electronegative fluorine and $Ca^{2+}$ cations are more electronegative than the organic PDMS polymers (Supplementary Note 12). Based on the above model, the possible ML process of the $CaF_2$:$Tb^{3+}$/PDMS elastomer is depicted in Fig. 3d, which shows that before being compounded, the potential wells of $CaF_2$:$Tb^{3+}$ and PDMS are completely separated (I)[30]. However, the phosphor and PDMS are in close contact after being compounded, and the two single potential wells become an asymmetric double potential well. Consequently, the energy barrier between the two wells is lowered. Then, even if there is no friction between the phosphors and the PDMS, the electrons at higher levels transfer from the PDMS to the phosphor to maintain energy level balance (II) (Supplementary Note 13). When the elastomer is stretched to reach a critical stretching distance, the previously contacted surfaces of the phosphors and PDMS are slightly separated. The electrons remain on the surface of the phosphors, resulting in negative electrostatic charges (electrons) on the phosphors and

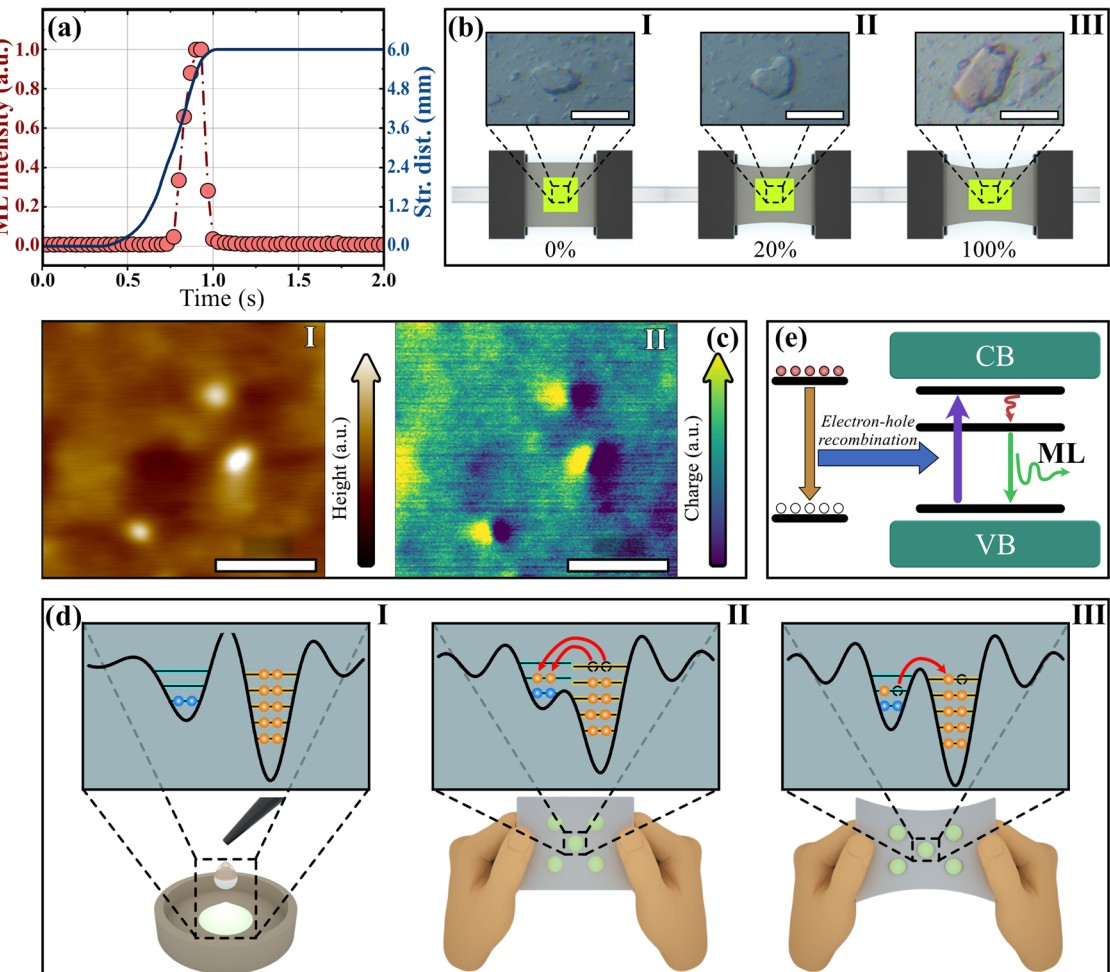

**Fig. 3 | Contact–separation–induced ML mechanism of the CaF$_2$:Tb$^{3+}$/PDMS elastomer. a** ML intensity and stretching distance of the elastomer during a single stretching (the "a. u." is the abbreviation of "arbitrary units"; source data are provided as a Source Data file); **b** optical microscopy images of the elastomer before stretching (I) and after stretching for 20% (II), 100% (III) of the maximum stretching distance (the scale bars in the inset are 200 μm); **c** AFM (I) and EFM (II) of the elastomer (the scale bars are 5 μm; the color scale arrows represent the relative height and charge density, respectively; the "a. u." is the abbreviation of "arbitrary units"); **d** single potential wells of the phosphor and PDMS before being compounded (I), double potential wells of the phosphor and PDMS in the elastomer before stretching (II) and under stretching (III): the black curves represent the

potential wells; the blue/orange circles on the lines represent electrons on levels in the potential wells; the red arrows indicate the electron transfer between the potential wells; **e** contact–separation–induced direct excitation of Tb$^{3+}$ emitters (the CB/VB are the acronyms of conduction band/valence band; the red and white circles represent the electrons and holes, respectively; the yellow and blue arrows indicate the electron–hole recombination and energy transfer to the emitters, respectively; the purple, red, and green arrows indicate the excitation, energy relaxation, and emission of the emitters, respectively). AFM, atomic force microscope; EFM electrostatic force microscopy; ML mechanoluminescence; PDMS polydimethylsiloxane.

positive electrostatic charges (holes) on the PDMS due to contact electrification. At the same time, the electron–hole pairs generate a strong electrostatic field. Because the gaps between the phosphors and the PDMS are very tiny, the distance between the electrons and the holes is also short. Therefore, the negative electrons on the surface of the phosphors can be attracted back to the PDMS by the opposite charge in a short time (III). Consequently, the electron–hole recombination occurs at the phosphor–PDMS interface to release excitation energy, thereby exciting the nearby Tb$^{3+}$ emitters for ML, as shown in Fig. 3e. In our lives, this electron–hole recombination generally occurs to induce some interesting static electricity phenomena, such as static sparks and beeping noise. For a typical example, we may see the bright static sparks when we take off a sweater on a dry winter night. (Please refer to Supplementary Note 14 for the effect of humidity and other factors on the ML intensity.) In addition, although the scratching stress applied to the CaF$_2$:Tb$^{3+}$/PDMS elastomer is small, the local pressure on the elastomer is very high, which is sufficient to induce the tiny interface separation

between the phosphors and the PDMS. As a result, it seems that we cannot "see" the scratching-induced separation between the phosphors and the PDMS with our naked eyes, it does occur. (Please refer to Supplementary Note 15 for more explanation on scratching-induced separation.)

At this stage, the stable self-recoverable ML of the CaF$_2$:Tb$^{3+}$/PDMS elastomer can be described based on contact electrification. According to contact electrification, the charge is transferred during contact and produced during separation[65–67]. Therefore, the ML self-recoverability of the centrosymmetric-phosphor/PDMS elastomer should depend on the stability of the contact–separation cycle at the phosphor–PDMS interface[30,31]. Accordingly, a contact–separation cycle model at the interface of centrosymmetric-phosphor/PDMS is established, as shown in Fig. 4a, and the energies of each state in the contact–separation cycle are calculated (Supplementary Note 16, 17)[68–72]. For comparison, the state energies of the typical centrosymmetric-phosphor-based Lu$_3$Al$_5$O$_{12}$–PDMS and Y$_2$O$_2$S–PDMS couple are also calculated. Figure 4b depicts the state energies in the

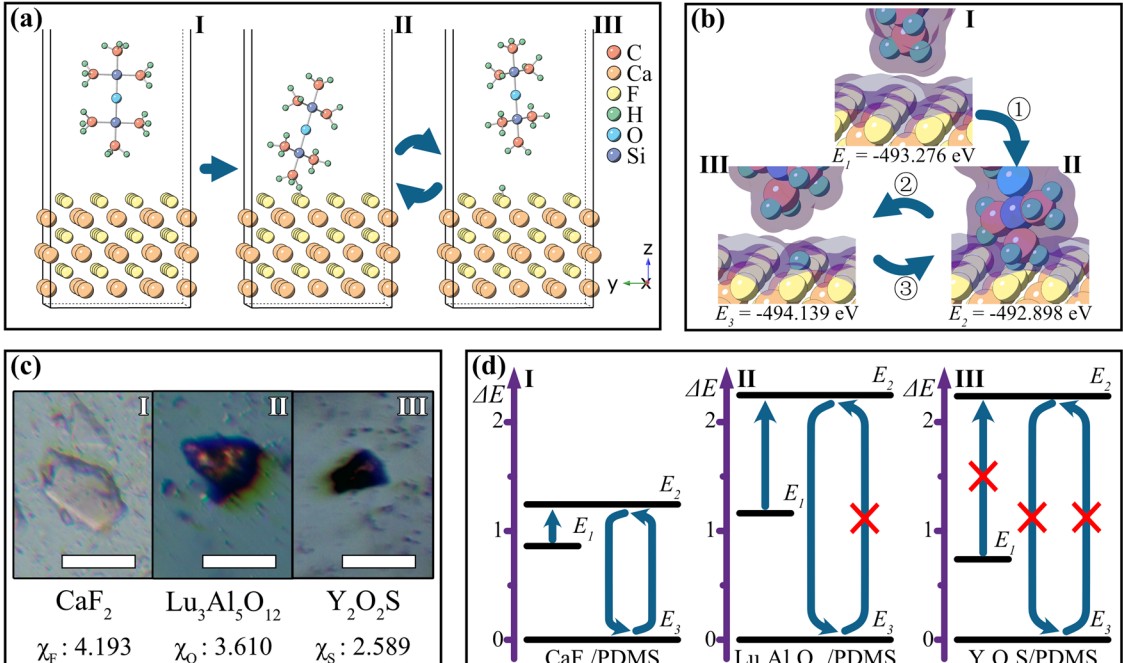

**Fig. 4 | Understanding of the self-recoverable ML mechanism of the CaF₂:Tb³⁺/ PDMS elastomer based on a contact–separation cycle model.**
**a** Contact–separation cycle models of the CaF₂–PDMS couple at the initial state (I), the contact state (II), and the separation state (III) (the blue arrows indicate the direction and pathway of the contact–separation cycle); **b** state energies (E₁, E₂, and E₃) of the CaF₂–PDMS couple at the initial state (I), the contact state (II), and the separation state (III) (the blue arrows indicate the direction and pathway of the energy-state cycle; the purple shades indicate the range of the atomic interactions;

**c** electronegativity (χ, using the Allen scale[73]) of F, S, and O and optical microscopy images of the CaF₂:Tb³⁺/PDMS (I), Lu₃Al₅O₁₂:Ce³⁺/PDMS (II), and Y₂O₂S:Eu³⁺/PDMS (III) elastomers after hard stretching three times (the scale bars in the inset are 200 μm); **d** contact–separation cycles of the CaF₂–PDMS (I), Lu₃Al₅O₁₂–PDMS (II), and Y₂O₂S–PDMS (III) couples (the black lines represent the energy states (E₁, E₂, and E₃); the blue arrows indicate the direction and pathway of the contact–separation cycle; the red crosses indicate that the pathways are not allowed). ML mechanoluminescence; PDMS polydimethylsiloxane.

contact–separation cycle, which shows that before being compounded into the PDMS, CaF₂ does not contact the PDMS with the energy $E_1 = -493.276$ eV at the initial state (I). After being compounded, CaF₂ and PDMS closely contact each other in the elastomer, so electrons transfer from PDMS to CaF₂ due to contact electrification. The coupled system rises to contact state (II), and thus, the state energy increases to $E_2 = -492.898$ eV. Correspondingly, Fig. 4d (I) indicates that the CaF₂–PDMS couple requires the energy $\Delta E_{21} = 0.378$ eV from the initial state to the contact state. When the elastomer is stretched beyond the critical distance, CaF₂ and PDMS in close contact are separated in a short time, resulting in rapid electron–hole recombination. Accordingly, the coupled system drops to the separation state (III), and the state energy decreases to $E_3 = -494.139$ eV. Subsequently, CaF₂ and PDMS come into contact again after stretching due to contraction of the elastomer, and the CaF₂–PDMS couple returns to contact state (II). When the elastomer is stretched again, the next contact–separation cycle of the CaF₂–PDMS interface begins. Figure 4d (I) indicates that the energy $\Delta E_{23} = 1.241$ eV is required to maintain the following contact–separation cycle at the interface of the CaF₂–PDMS couple.

Finally, we compare the state energies of the CaF₂–PDMS, Lu₃Al₅O₁₂–PDMS, and Y₂O₂S–PDMS stacks in Fig. 4d. The $\Delta E_{21}$ represents the energy difference between the initial state $E_1$ and the contact state $E_2$ and can be used to evaluate the number of electrons transferred from PDMS to phosphors due to the contact electrification effect. Accordingly, the $\Delta E_{21}$ energy of the Y₂O₂S–PDMS couple (1.502 eV) is clearly larger than the others, indicating that it is more difficult to induce contact electrification at the Y₂O₂S–PDMS interface. Consequently, we barely observe ML for the centrosymmetric-phosphor-based Y₂O₂S:Eu³⁺/PDMS elastomer even under intense stretching, as shown in Fig. 2i. The $\Delta E_{23}$ represents the energy

difference between the contact state $E_2$ and the separation state $E_3$, and therefore it can be applied to evaluate the self-recovery ability of the contact–separation cycle. However, since the contact electrification of the Y₂O₂S–PDMS couple is weak, it is meaningless to discuss its ML self-recovery ability due to contact electrification. In other words, as for the Y₂O₂S–PDMS couple, it cannot rise from $E_1$ to $E_2$ state, and therefore it can never drop to $E_3$ state. Moreover, the energy $\Delta E_{23}$ of the Lu₃Al₅O₁₂–PDMS couple (2.251 eV) is significantly larger than that of CaF₂–PDMS (1.241 eV), which correspondingly indicates that much more energy is required to maintain the contact–separation cycle at the Lu₃Al₅O₁₂–PDMS interface. This result suggests that it is more difficult to recover the surface contact of the Lu₃Al₅O₁₂–PDMS couple after separation. Correspondingly, Fig. 4c presents the microscopes of the CaF₂:Tb³⁺/PDMS (I), Lu₃Al₅O₁₂:Ce³⁺/PDMS (II), and Y₂O₂S:Eu³⁺/PDMS (III) elastomers after hard stretching three times. The Lu₃Al₅O₁₂:Ce³⁺/PDMS and Y₂O₂S:Eu³⁺/PDMS elastomers are significantly dyed dark red, demonstrating that their phosphor-PDMS interfaces have been seriously damaged during the stretching process. As a result, the Lu₃Al₅O₁₂:Ce³⁺/PDMS elastomer shows very poor ML self-recoverability, with complete ML degradation after only three stretches. In contrast, the energy $\Delta E_{23}$ (1.241 eV) at the CaF₂–PDMS interface is low enough that the CaF₂:Tb³⁺ phosphor can restore close contact after separation from the PDMS, resulting in significantly less damage at the interface and self-recoverable ML under stretching. Furthermore, the above results suggest that the fluoride–PDMS couple helps to induce more efficient contact electrification and maintain a more stable contact–separation cycle at the interfaces, so it would be a favorable strategy to develop stable self-recoverable ML elastomers based on centrosymmetric fluoride phosphors and PDMS polymers.

## Discussion

Here, we report a new type of $CaF_2$:$Tb^{3+}$/PDMS elastomer with a self-recoverable ML under stretching, which is demonstrated to be induced by contact electrification arising at the interface between the centrosymmetric $CaF_2$:$Tb^{3+}$ and the PDMS. According to the contact electrification model, we successfully establish a contact−separation cycle for the interface of the phosphor−PDMS couple. By first-principles calculations, the state energies in the contact−separation cycle are evaluated to understand the self-recoverable ML of the $CaF_2$:$Tb^{3+}$/PDMS elastomer under stretching. The results indicate that both $\Delta E_{21}$ and $\Delta E_{23}$ of the $CaF_2$−PDMS couple are sufficiently low, which reveals that the fluoride−PDMS couple helps to maintain a more stable contact−separation cycle at the interface. This effect results in the self-recoverable ML of the $CaF_2$:$Tb^{3+}$/PDMS elastomer. Accordingly, we propose developing self-recoverable ML elastomers based on centrosymmetric fluoride phosphors and PDMS.

## Methods

### Synthesis of the $CaF_2$:$Tb^{3+}$

The $CaF_2$:$Tb^{3+}$ phosphor was synthesized using the conventional solid-phase method. The raw materials utilized were $CaF_2$ (99.99%) and $TbF_3$ (99.99%). Initially, the raw materials were weighed according to the required proportions and ground and mixed with an agate mortar. The mixture was then transferred into an alumina crucible, which was placed in a large porcelain crucible with a lid containing carbon powder. The crucible was subsequently heated at 1250 °C for 6 h in air in a muffle furnace. Once the synthesis was complete, the sample was allowed to cool to room temperature before being ground again with an agate mortar to produce the $CaF_2$:$Tb^{3+}$ phosphor.

### Preparation of ML elastomers

The ML elastomer used PDMS of Sylgard 184 type from Dow Corning as its matrix. The PDMS base resin and curing agent were mixed in a 2 ml medical syringe at a ratio of 10:1. The phosphor was added to the mixture at a mass ratio of 1 ($CaF_2$:$Tb^{3+}$) to 1 (PDMS) and stirred evenly. The resulting mixture was poured into a Petri dish and solidified at 60 °C for 3 h in an oven. After cooling, the sample was removed from the dish and cut into a square with a side length of 1 cm. The square was then placed back into the Petri dish and covered with PDMS, which was allowed to solidify. Finally, the sample was removed and cut into a specific shape for analysis and testing.

### Characterizations

The X-ray diffraction (XRD) patterns were measured using a Rigaku D/Max-2400 X-ray diffractometer, while PL spectra were obtained using an FLS-920T fluorescence spectrophotometer. Scanning electron microscopy (SEM) was used to observe the particle morphology and acquire the element distribution characteristics of the particles, with an FEI-Apreo S instrument. The surface morphology and surface electron distribution of the samples were obtained by using an Oxford Cypher S AFM Microscope (Supplementary Note 18). The valence states of the ions were identified through X-ray photoelectron spectroscopy (XPS) using the PHI-5702 model. Luminescence signals were collected in situ through an optical fiber connected to a collimator (BFC-441) and then transferred to an Omni-λ300i spectrometer equipped with a CCD camera (iVac-316). Photos and videos were captured with a Huawei phone (P40 Pro). Finite element simulations were carried out using the Caelinux 2018 operating system integrated with Salome MECA 2018.

### First-principles calculation

First-principles calculations are performed by Vienna ab initio simulation package (VASP)[68–72]. The generalized gradient approximation(GGA) of Perdew−Burke−Ernzerhof (PBE) is used to describe the exchange-correlation function. The cut-off energy for the plane wave basis is set to 400 eV and a $2 \times 2 \times 1$ mesh is employed. All the structures were fully relaxed (atomic position) up to $10^{-5}$ eV /Å force minimization and max force of 0.01 eV/Å.

### Reporting summary

Further information on research design is available in the Nature Portfolio Reporting Summary linked to this article.

## Data availability

Source data are provided with this paper in the Figshare database, ref. 74. The SI data are available from the corresponding author upon request. Source data are provided with this paper.

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

## Acknowledgements
This work was supported by the National Natural Science Foundation of China (No. 10904057 and 12074159 to J.Z) and the Science and Technology Projects of Gansu Province (No. 18JR3RA270 to J.Z).

## Author contributions
W. Wang and J. Zhang conceived the ideas and designed the research. W. Wang, J. Zhang, and S. Wang established the theoretical model. W. Wang and S. Wang performed the experiments. W. Wang, Y. Gu, and J. Zhou. analyzed the data. W. Wang, S. Wang, Y. Gu, J. Zhou, and J. Zhang interpreted the results and wrote the manuscript with input from all authors. J. Zhang supervised the study.

## Competing interests
The authors declare no competing interests.
