## [Peer Review File · Nature Communications]

Contact-separation-induced self-recoverable
mechanoluminescence of CaF₂:Tb³⁺/PDMS elastomerREVIEWER COMMENTS

Reviewer #1 (Remarks to the Author):

In this manuscript, authors report the significant self-recoverable ML of CaF₂:Tb³⁺/PDMS elastomer under stretching and proposed a contact-separation cycle model to understand the self-recoverable ML well. The work presents interesting and enlightening results, and the experiments as well as the calculations are well conducted. The supplementary files attached are also highly informative and impressive. Therefore, I can recommend it for publication after fully consideration of the following issues.

1. In Figure 2, it describes that the ML and PL follow the same luminescent pathway, however their spectra are still different. This spectral difference should be explained.
2. When Tb³⁺ ions are doped in CaF₂, how to keep the charge balance? Please discuss it.
3. In Figure 3(e), it describes that the ML is stimulated by a strange process labelled "electronic composite". I can't agree with this description, and I believe it may be due to the carrier recombination based on the results in the Figure 3(a-c). The author should check it carefully.
4. In Figure 4(c), the electronegativity of S should be 2.58 rather than 2.4 according to the Periodic table of elements. Please revise it.
5. In Figure 4(d), it can drop to E₃ state for a larger energy (2.25 eV) of the Lu₃Al₅O₁₂/PDMS, however it can't drop to E₃ state for a lower energy (2.23 eV) of the Y₂O₂S/PDMS. The author should provide detailed and reasonable explanations.
6. The results in Supplementary Figure S2-13 (falling ball experiment) are also significant for demonstrating the ML self-recoverability of the CaF₂:Tb³⁺/PDMS elastomer under continuous slight strain, and therefore should be described in detail. However, I can't find any description about the falling ball experiment in the Results and discussion section.
7. In Supplementary Note 3, the 2D stretching machine in Figure S3 is obviously different from the machine in Figure 2(f). The author should carefully examine and explain them.

Reviewer #2 (Remarks to the Author):

In this paper, the authors have demonstrated self-recoverable mechanoluminescence (ML) using CaF₂:Tb³⁺/PDMS without the need for pre-irradiation. They assert that their finding represents a unique and promising phenomenon, as previous attempts with centrosymmetric-oxide/PDMS-induced ML exhibited weak emission and significant degradation under stretching conditions. The authors propose that this "partially" self-recoverable behavior observed in CaF₂:Tb³⁺/PDMS is induced by contact-separation-induced electrification. The development of new ML materials with self-recoverable ML behavior is important in the field of ML and matches well with the broad scope of this journal. The results presented are substantial and support their claims. Therefore, I recommend the publication of this manuscript; however, several issues need to be addressed before publication.

1. In the "In this work...." section, the authors used the phrase "partially self-recover." Is this phrase used consciously to describe the slight decrease in ML intensity observed in Figure 2(h)? I recommend providing a more explicit explanation and additional context, in the introduction, to help readers better understand this concept.
2. Following the previous comment, the authors also employed the terms "slight scratching" and "hard stretching," which are very unclear in meaning. The authors need to explain why scratching and stretching are different in the viewpoint of "slight" and "hard".
3. The results presented in this paper show an improvement compared to the authors' previous work (ref 24), which was susceptible to stretching-induced damage. The previous results were reported to suffer from damage caused by stretching, whereas CaF₂:Tb³⁺/PDMS exhibits less damage and suggests a self-recovery property. However, the term "damage" is somewhat ambiguous in this context. It is necessary to provide a more intuitive description and explanation of what constitutes "damage."

4. To obtain the results presented in Figure 2(h), the authors claim to have avoided pre-irradiation with UV or blue light. However, it is possible that fluorescent or sunlight exposure could have occurred during synthesis, sample preparation, and measurement. These light sources also contain UV and blue light components. How did the authors account for and rule out the potential influence of these sources?

5. The authors contend that CaF₂:Tb³⁺ is more electronegative, leading to electron attraction from PDMS. However, according to the triboelectric series, PDMS is already relatively electronegative. Strategically, it might be more effective in terms of increasing contact electrification to make the ML particles more electropositive. Please provide the authors' perspective on this matter.

6. In the explanation of the ML process, it is mentioned that after critical stretching, electrons transfer to CaF₂:Tb³⁺ and subsequently return to PDMS, causing direct excitation. What is the mechanism behind the return of electrons to PDMS? Is it due to breakdown or neutralization? Since the returned electrons contribute to ML, it is important to provide a detailed explanation of this phenomenon.

7. The term "self-recovery" was originally introduced by Chandra (ref. 60) to explain the first demonstration of 100,000 MLs in the ZnS:Cu/PDMS system, even without UV/blue light pretreatment (Appl. Phys. Lett. 102, 051110 (2013)). If the authors' results emphasize the significance of this aspect, it would be beneficial to include a brief history of self-recoverable ML in the introduction to aid readers' comprehension.

Reviewer #3 (Remarks to the Author):

This manuscript reports the self-recoverable ML elastomers based on centrosymmetric fluoride phosphors and PDMS. Contact electrification is deemed to be responsible for the ML mechanism, where a contact-separation model is proposed to explain the detailed phenomena. However, the similar contact electrification induced ML in several recently-developed phosphors has been reported by the same group previously (Nano Energy 94 (2022) 106920, et.al.,). Although the authors have argued the major difference (i.e., the ML resuming ratio after each hard stretching) between the present ML phenomenon and the previous ones, it is hard to convince one that this difference is so significant that deserves to be published in Nature Communications. In addition, some issues need to be considered by the authors as follows.

1.The authors keep emphasizing the ML comes from the centrosymmetric structure. Why is this so important? In my opinion, the contact electrification induced ML may happen for phosphors with different structures, as long as the contact electrification occurs.

2.According to the authors, the separation between the phosphor and PDMS is important for the ML, since the contact-electrification induced electrons are transferred to the phosphor when the CaF₂-PDMS interface is separated. If this is true, how to explain the ML from scratching? I don't see that there would be any separation between CaF₂ and PDMS when scratching happens.

3.What technique has been adopted to acquire the surface morphology and surface electron distribution in Fig.3C?

4.The authors argue that electrons transfer from PDMS to CaF₂ as long as the CaF₂ and PDMS are compounded. To this point, is the friction not necessary for the triboelectrification?

5.According to the authors, the rapid electron-hole recombination happens only after the separation between CaF₂ and PDMS. If the electrons transfer from the PDMS to CaF₂, holes should be left in PDMS. In this regard, I cannot understand how the electron-hole recombination may happen when electrons and holes are spatially separated.

In general, I think the contact-separation model is not convincing. Therefore, I cannot recommend the publication of this manuscript.

RESPONSE TO REVIEWERS' COMMENTS

Reviewer #1 (Remarks to the Author):

In this manuscript, authors report the significant self-recoverable ML of $\text{CaF}_2:\text{Tb}^{3+}/\text{PDMS}$ elastomer under stretching and proposed a contact-separation cycle model to understand the self-recoverable ML well. The work presents interesting and enlightening results, and the experiments as well as the calculations are well conducted. The supplementary files attached are also highly informative and impressive. Therefore, I can recommend it for publication after fully consideration of the following issues.

1. In Figure 2, it describes that the ML and PL follow the same luminescent pathway, however their spectra are still different. This spectral difference should be explained.

Response: dear reviewer, thank you very much for pointing out this confusing issue in Figure 2 and we apologize for not explaining this issue in the original manuscript. As you mentioned in the comments, the peak widths of the ML and PL emission spectra are clearly different in Figure 2a, and the width of the ML spectrum is significantly broader than that of the PL spectrum. In fact, this difference should be due to the different spectral measurement equipment. The PL of the $\text{CaF}_2:\text{Tb}^{3+}/\text{PDMS}$ elastomer is continuous under irradiation and can be recorded using a general FLS-920T fluorescence spectrophotometer. However, the ML of the elastomer is instantaneous under stress, so it can be only recorded using a CCD camera. In this case, the ML signals were collected in situ through an optical fiber connected to a collimator (BFC-441), and then transferred to an Omni- λ 300i spectrometer equipped with a CCD camera (iVac-316). Since the ML is always relatively weaker than the PL, the CCD slit for the ML is significantly larger than that of the PL, resulting in the significantly broader spectral peaks. If the same slits are applied in spectral measurement, we can only see the PL peaks and the details of the weaker ML will be completely lost in the strong PL. Moreover, although the peak widths of the ML and PL emission spectra show some differences, we can only observe the characteristics $^5\text{D}_4 \rightarrow ^7\text{F}_j$ emissions and cannot observe the blue emissions originating from the $^5\text{D}_3 \rightarrow ^7\text{F}_j$ transitions. Therefore, it can be still concluded that the Tb^{3+} emitters should be excited directly rather than through excitation relaxation from conduction band similar to CL (for example electron bombardment). Accordingly, it indicates that the ML and PL follow the similar

emission paths, resulting in the similar emission colors as shown in Figure 2c. According to your significant comments, we have revised our manuscript and explained this issue in detail in the revision.

2. When Tb^{3+} ions are doped in CaF_2 , how to keep charge balance? Please discuss it.

Response: dear reviewer, thank you very much for the significant comments. As you commented, the Tb^{3+} ion is trivalent, but there is only one kind of bivalent calcium ion in the CaF_2 crystal, so if the Tb^{3+} replaces the Ca^{2+} site, it will inevitably induce a significant charge imbalance. Generally, there are two possible methods to keep the charge balance when the Tb^{3+} ions are doped in the CaF_2 crystal. First, two Tb^{3+} ions replace three Ca^{2+} cations, generating one Ca^{2+} vacancy. Second, one Tb^{3+} ion replace two Ca^{2+} cations, and it generates one Ca^{2+} vacancy and one F^- vacancy.

To figure out the charge compensation mechanism of the non-equivalent doping of Tb^{3+} in CaF_2 crystal, three specific kinds of defects, which are $[Tb_{Ca}^{\bullet}]$, Ca^{2+} vacancy $[V_{Ca}^{\bullet}]$ and F^- vacancy $[V_F^{\bullet}]$ were built up in CaF_2 crystal as shown in the following Figure R1. Then, the DFT calculations based on a VASP package are conducted. Finally, the cohesive energies for the different defects including the $[Tb_{Ca}^{\bullet}]$ - $[V_F^{\bullet}]$ - $[V_{Ca}^{\bullet}]$, $2[Tb_{Ca}^{\bullet}]$ - $[V_{Ca}^{\bullet}]$ and $[Tb_{Ca}^{\bullet}]$ can be evaluated and the DFT calculation results are presented in the Table R1.

Figure R1. Three specific kinds of defects including $[Tb_{Ca}^{\bullet}]$, Ca vacancy $[V_{Ca}^{\bullet}]$ and F vacancy $[V_F^{\bullet}]$ in CaF_2 crystal for DFT calculation, and the Tb^{3+} substitution ratio is unified to 2/32.

Table R1. DFT calculated cohesive energies (E_c) for the different defects including

the $[\text{Tb}_{\text{Ca}}\cdot]-[\text{V}_{\text{F}}\cdot]-[\text{V}_{\text{Ca}}\cdot]$, $2[\text{Tb}_{\text{Ca}}\cdot]-[\text{V}_{\text{Ca}}\cdot]$ and $[\text{Tb}_{\text{Ca}}\cdot]$ in the CaF_2 crystal.

Defects	Number of atoms	Cell E_c (eV)	Atomic E_a (eV)
$[\text{Tb}_{\text{Ca}}\cdot]-[\text{V}_{\text{F}}\cdot]-[\text{V}_{\text{Ca}}\cdot]$	92	-536.125	-5.796
$2[\text{Tb}_{\text{Ca}}\cdot]-[\text{V}_{\text{Ca}}\cdot]$	95	-556.630	-5.828
$[\text{Tb}_{\text{Ca}}\cdot]$	96	-557.806	-5.779

Table R1 presents the cell cohesive energies (E_c) for the different defects in the CaF_2 crystal. Since there are different number of atoms in these cells, the cell E_c should be subtracted from the free state energy of each atom and divided by the number of atoms to obtain the atomic E_a for different defects. The results indicate that the atomic E_a (-5.828 eV) for the $2[\text{Tb}_{\text{Ca}}\cdot]-[\text{V}_{\text{Ca}}\cdot]$ defect cluster is the smallest as shown in Table R1. It means that two Tb^{3+} dopants would replace three Ca^{2+} cations in the CaF_2 crystal, resulting in one additional Ca^{2+} vacancy $[\text{V}_{\text{Ca}}\cdot]$ to keep the charge balance in the CaF_2 crystal.

However, the above discussion based on the DFT calculations is still a theoretical analysis, and at present there is still a lack of sound experimental evidence for which defect belongs to. Here, we can only make some theoretical discussions to help the readers understand the charge compensation mechanism of the non-equivalent doping of Tb^{3+} in CaF_2 crystal. In response to your significant comments, we have carefully revised the manuscript, and the discussions about the charge compensation mechanism of the non-equivalent doping of Tb^{3+} in CaF_2 crystal have been appropriately added in the revision. Thank you very much for your significant comments and constructive suggestion again.

3. In Figure 3(e), it describes that the ML is stimulated by a strange process labelled “electronic composite”. I can’t agree with this description, and I believe it may be due to the carrier recombination based on the results in the Figure 3(a-c). The author should check it carefully.

Response: dear reviewer, thank you very much for pointing out this error in the Figure 3(e) and we are very sorry for such an undeserved error in the manuscript. As you commented, the direct excitation energy of the Tb^{3+} emitters should be due

to the recombination of electrons and holes, i.e., the recombination of the charge carriers, rather than the “electronic composite”. According to your comments, we have corrected this wrong description in the Figure 3(e) and this corrected figure has been presented in the response letter as following Figure R2.

Figure R2. Corrected Figure 3(e): ML process of the $\text{CaF}_2:\text{Tb}^{3+}/\text{PDMS}$ elastomer.

4. In Figure 4(c), the electronegativity of S should be 2.58 rather than 2.4 according to the Periodic table of elements. Please revise it.

Response: dear reviewer, thank you very much for pointing out this error. According to your comments and the standard periodic table of elements, we have carefully examined and revised the electronegativity data in the Figure 4(c). The electronegativity of the S, O, F have been accurately corrected to 2.589, 3.610 and 4.193 in the revision, respectively. The corrected figure has been presented in the response letter as following Figure R3.

Figure R3. Corrected Figure 4(c): electronegativity of the F, S, O and microscopes images of the $\text{CaF}_2:\text{Tb}^{3+}/\text{PDMS}$, $\text{Lu}_3\text{Al}_5\text{O}_{12}:\text{Ce}^{3+}/\text{PDMS}$ and $\text{Y}_2\text{O}_2\text{S}:\text{Eu}^{3+}/\text{PDMS}$ elastomers after hard stretching 3 times (inset).

5. In Figure 4(d), it can drop to E_3 state for a larger energy (2.25 eV) of the $\text{Lu}_3\text{Al}_5\text{O}_{12}/\text{PDMS}$, however it can't drop to E_3 state for a lower energy (2.23 eV) of the $\text{Y}_2\text{O}_2\text{S}/\text{PDMS}$. The author should provide detailed and reasonable explanations.

Response: dear reviewer, thank you very much for the significant comments and we are very sorry that we did not explain this issue clearly. According to your comments, we have revised our manuscript carefully and have provided a detailed and reasonable explanation in the revision. To understand the proposed contact-separation cycle in detail, it is necessary to first clarify the definitions of the ΔE_{21} and ΔE_{23} . Based on the contact-separation cycle model at the interface of centrosymmetric-phosphor/PDMS, the ΔE_{21} represents the energy difference between the initial state E_1 and contact state E_2 and can be used to evaluate the number of electrons transferred from PDMS to phosphor due to the contact electrification effect. According to the calculation results in Figure 4(d), the ΔE_{21} of $\text{Y}_2\text{O}_2\text{S}/\text{PDMS}$ (1.502 eV) is larger than that of $\text{Lu}_3\text{Al}_5\text{O}_{12}\text{-PDMS}$ (1.083 eV) and is significantly larger than that of $\text{CaF}_2\text{-PDMS}$ (0.378 eV). Correspondingly, it was experimentally demonstrated that the $\text{Y}_2\text{O}_2\text{S}:\text{Eu}^{3+}/\text{PDMS}$ elastomer cannot emit any ML even under strong stretching. ML of $\text{Lu}_3\text{Al}_5\text{O}_{12}:\text{Ce}^{3+}/\text{PDMS}$ elastomer can

be clearly observed under stretching, but it is significantly weaker than that of $\text{Y}_2\text{O}_2\text{S}:\text{Eu}^{3+}/\text{PDMS}$ elastomer. Based on the above calculation and experimental results, it can be concluded that the larger ΔE_{21} is, the weaker contact electrification effect is and the weaker ML is. Furthermore, the ΔE_{23} represents the energy difference between the contact state E_2 and separation state E_3 , and therefore it can be applied to evaluate the self-recovery ability of the contact-separation cycle. However, since the contact electrification of $\text{Y}_2\text{O}_2\text{S}-\text{PDMS}$ couple is so weak that ML cannot be excited under stress, it is meaningless to discuss its self-recovery ability due to contact electrification. In other words, as for the $\text{Y}_2\text{O}_2\text{S}-\text{PDMS}$ couple, since it cannot rise from E_1 to E_2 state, therefore it can never drop to E_3 state. As for $\text{Lu}_3\text{Al}_5\text{O}_{12}-\text{PDMS}$ couple, although its ΔE_{21} is also large, the experiments demonstrate that the $\text{Lu}_3\text{Al}_5\text{O}_{12}:\text{Ce}^{3+}/\text{PDMS}$ elastomer shows ML under stretching. The result reveals that the $\text{Lu}_3\text{Al}_5\text{O}_{12}-\text{PDMS}$ couple can still rise to E_2 state and thus it can drop to E_3 state as well. However, since the ΔE_{23} of $\text{Lu}_3\text{Al}_5\text{O}_{12}-\text{PDMS}$ (2.251 eV) is clearly larger than that of CaF_2-PDMS (1.241 eV), it is more difficult for the $\text{Lu}_3\text{Al}_5\text{O}_{12}-\text{PDMS}$ couple to recover E_2 state well. Correspondingly, the $\text{Lu}_3\text{Al}_5\text{O}_{12}:\text{Ce}^{3+}/\text{PDMS}$ elastomer shows severe ML degradation due to the poor self-recovery ability, and thus its ML will significantly decrease to almost zero after being stretched only three times.

6. The results in Supplementary Figure S2-13 (falling ball experiment) are also significant for demonstrating the ML self-recoverability of the $\text{CaF}_2:\text{Tb}^{3+}/\text{PDMS}$ elastomer under continuous slight strain, and therefore should be described in detail. However, I can't find any description about the falling ball experiment in the Results and discussion section.

Response: dear reviewer, thank you very much for your comments and this is a very constructive suggestion. According to your significant suggestion, we have revised the original draft and added some discussions about the falling ball experiment in the revision. Also, the falling ball experimental device and results have been provided in the Figure 2 corresponding to Figure R4 in the response letter. It shows that the $\text{CaF}_2:\text{Tb}^{3+}/\text{PDMS}$ elastomer is fixed on a transparent platform, and then a 15.0 g iron ball is transported to the top of the elastomer with an electromagnet. The

distance of the ball to the platform is fixed at 0.5 m. Finally, the iron ball falls free and hits the elastomer to excite bright green ML as shown in Figure 2f(i). At this time, the ML signal will be recorded by a CCD camera under the transparent platform. Figure 2f(ii) indicates that the ML of the $\text{CaF}_2:\text{Tb}^{3+}/\text{PDMS}$ elastomer can self-recover well under continuous hitting, and it can keep stable even after being hit 50 times, demonstrating excellent self-recoverability of this elastomer under continuous slight hitting.

Figure R4. Corrected Figure 2(f): a falling ball experimental device to test ML of the elastomer under hitting: ML photo (i) and ML intensities under hitting (ii).

7. In Supplementary Note 3, the 2D stretching machine in Figure S3 is obviously different from the machine in Figure 2(f). The author should carefully examine and explain them.

Response: dear reviewer, thank you very much for pointing out this potential confusing issue and we sincerely apologize for incorrectly using the initial version of the 2D stretching machine diagram in the Figure 2(f). At the beginning of the 2D stretching experiment, the first version of the 2D stretching machine in the original Figure 2(f) was used to measure the ML properties of the $\text{CaF}_2:\text{Tb}^{3+}/\text{PDMS}$

elastomer under continuous stretching. However, since its stretching frequency was too high, the stretching stress was insufficient, resulting in a weak ML. Therefore, in order to control the stretching frequency of the 2D stretching machine, we added two gearshift-based planetary reducers to the 2D stretching machine, as shown in the current Figure 2(g). Consequently, the stretching speed, distance and frequency of the 2D stretching machine can be accurately controlled at about 10.0 mm/s, 6.0 mm and 1.0 Hz. According to your significant comments, we have corrected the original Figure 2(f) to the current Figure 2(g) in the revision and the corrected figure is also presented in the following Figure R5 in the response letter.

Figure R5. Corrected Figure 2(g): a two-dimensional stretching machine to test ML of the elastomer under continuous stretching.

Reviewer #2 (Remarks to the Author):

In this paper, the authors have demonstrated self-recoverable mechanoluminescence (ML) using $\text{CaF}_2:\text{Tb}^{3+}/\text{PDMS}$ without the need for pre-irradiation. They assert that their finding represents a unique and promising phenomenon, as previous attempts with centrosymmetric-oxide/PDMS-induced ML exhibited weak emission and significant degradation under stretching conditions. The authors propose that this "partially" self-recoverable behavior observed in $\text{CaF}_2:\text{Tb}^{3+}/\text{PDMS}$ is induced by contact-separation-induced electrification. The development of new ML materials with self-recoverable ML behavior is important in the field of ML and matches well with the broad scope of this journal. The results presented are substantial and support their claims. Therefore, I recommend the publication of this manuscript; however, several issues need to be addressed before publication.

1. In the "In this work...." section, the authors used the phrase "partially self-recover." Is this phrase used consciously to describe the slight decrease in ML intensity observed in Figure 2(h)? I recommend providing a more explicit explanation and additional context, in the introduction, to help readers better understand this concept.

Response: dear reviewer, thank you very much for pointing out the significant issue in the introduction section and we sincerely apologize for not explaining this confusing issue. As you commented, the phrase "partially self-recover" is actually used to describe the less decrease in ML intensity observed in Figure 2(h). It means that although the new $\text{CaF}_2:\text{Tb}^{3+}/\text{PDMS}$ elastomer still shows some ML decrease after each stretching, it is very exciting to find for the first time that its ML can self-recover to approximately 60-75% of its ML intensity at previous stretching. Significantly, the bright green ML of the new $\text{CaF}_2:\text{Tb}^{3+}/\text{PDMS}$ elastomer can still be clearly observed even after being stretched vigorously more than 30 times. As a typical competing material, the ML of the previously reported optimal $\text{Lu}_3\text{Al}_5\text{O}_{12}:\text{Ce}^{3+}/\text{PDMS}$ elastomer cannot recover at all after stretching. Consequently, even if the $\text{Lu}_3\text{Al}_5\text{O}_{12}:\text{Ce}^{3+}/\text{PDMS}$ elastomer is only stretched 3 times, its ML intensity is drastically reduced to almost zero and its green ML is no longer observed. Accordingly, although the new $\text{CaF}_2:\text{Tb}^{3+}/\text{PDMS}$ elastomer still needs further improvement, this is the first time we have observed a stable

stretching-induced self-recoverable ML in a centrosymmetric-phosphor/PDMS elastomer. Therefore, it is very valuable to study in detail the self-recoverable ML mechanism of the $\text{CaF}_2:\text{Tb}^{3+}$ /PDMS elastomer for further improvement.

However, we deeply apologize for not providing a more explicit explanation on the confusing phrase "partially self-recover", which has caused difficulties in understanding. According to your constructive suggestion, we have revised the introduction section of our manuscript and have provided some explanation for this confusing phrase to help readers better understand the concept of "partially self-recover". Thank you very much for your good suggestion.

2. Following the previous comment, the authors also employed the terms "slight scratching" and "hard stretching," which are very unclear in meaning. The authors need to explain why scratching and stretching are different in the viewpoint of "slight" and "hard".

Response: dear reviewer, this is a very significant comment and thank you very much for pointing out these unclear expressions in our manuscript that easily lead to confusion in understanding. In this case, ML can be clearly observed when the pressure applied to the $\text{CaF}_2:\text{Tb}^{3+}$ /PDMS elastomer is greater than 0.2 MPa. Generally, the entire elastomer is stressed under stretching, while only a very tiny area in contact with the nail is stressed under scratching. Therefore, the stress-bearing area of the elastomer under stretching is much larger than that under scratching. Consequently, even if the same pressure (P) is applied, the stress ($F = P \times S$) under stretching must be much greater than the stress under scratching due to the much larger stress-bearing area (S). Correspondingly, Table R2 shows the stretching stresses and scratching stresses to the $\text{CaF}_2:\text{Tb}^{3+}$ /PDMS elastomer under different pressures. For example, it shows that when the pressure applied to the elastomer is 0.2341 MPa, the stretching stress (2.5571 N) is even more than 20 times the scratching stress (0.1176 N). Consequently, we always feel that the stress under scratching is very "slight" and the stress under stretching is very "hard". We are very sorry that we did not provide a detailed explanation for the difference between "slight" and "hard". According to your significant comments and constructive suggestion, we have carefully revised our manuscript and added some

appropriate explanation for this significant issue in the revision. Thank you very much for your significant comments and kind help again.

Table R2. Stretching stresses and scratching stresses to the CaF₂:Tb³⁺/PDMS elastomer under different pressures.

Pressure (MPa)	Stretching stress (N)	Scratching stress (N)
0.1261	1.3778	0.0634
0.2341	2.5571	0.1176
0.3304	3.6101	0.1660
0.4215	4.6051	0.2118
0.5074	5.5434	0.2549
0.5926	6.4743	0.2977
0.6798	7.4264	0.3415
0.7729	8.4442	0.3883
0.8799	9.6127	0.4421
1.0013	10.9390	0.5030

3. The results presented in this paper show an improvement compared to the authors' previous work (ref 24), which was susceptible to stretching-induced damage. The previous results were reported to suffer from damage caused by stretching, whereas CaF₂:Tb³⁺/PDMS exhibits less damage and suggests a self-recovery property. However, the term "damage" is somewhat ambiguous in this context. It is necessary to provide a more intuitive description and explanation of what constitutes "damage."

Response: dear reviewer, we sincerely apologize for not providing a more in-depth description and explanation of the ambiguity of term "damage" in the manuscript. This is a very constructive comment and thank you very much for pointing out this significant issue. According to your significant and constructive suggestion, we have carefully revised our manuscript and added a more intuitive description and explanation of what is "damage" in the revision.

Briefly, the term "damage" refers to the permanent tiny gaps generated at the

phosphor-PDMS interface that can be filled with red dyes, typically created by the separation of the phosphors and the PDMS during the contact-separation cycle. These tiny gaps at interface can hinder the next contact between the phosphors and the PDMS, thus significantly affecting the ML self-recoverability of the elastomer. In detail, the phosphor powders and the PDMS polymer in the elastomer are originally in close contact before stretching. When the elastomer is stretched, the phosphors and PDMS are briefly separated to induce ML. After stretching, the phosphors will come to contact with the PDMS again due to the natural contraction of the elastomer. Ideally, the phosphors and PDMS should return to the close contact they had before stretching. However, for the previously reported typical oxide phosphor such as $\text{Lu}_3\text{Al}_5\text{O}_{12}:\text{Ce}^{3+}$, it is actually difficult for the oxide phosphors and PDMS to recover close contact because more energy is required for the $\text{Lu}_3\text{Al}_5\text{O}_{12}$ -PDMS couple based on our DFT results. As a result, some tiny gaps inevitably arise between the oxide phosphors and the PDMS, defined in this case as the term "damage". Since the ML of the phosphor/PDMS elastomer is due to contact electrification at the interface of the phosphors and PDMS, these gaps will inevitably lead to insufficient contact between the phosphors and PDMS, which significantly decreases the ML of the elastomer at the next stretch.

4. To obtain the results presented in Figure 2(h), the authors claim to have avoided pre-irradiation with UV or blue light. However, it is possible that fluorescent or sunlight exposure could have occurred during synthesis, sample preparation, and measurement. These light sources also contain UV and blue light components. How did the authors account for and rule out the potential influence of these sources?

Response: dear reviewer, this is a very constructive comment and thank you very much for pointing out this significant issue in our manuscript. In general, for the previously reported long-persistent luminescence phosphors represented by $\text{SrAl}_2\text{O}_4:\text{Eu}^{2+}$, their PDMS-based elastomers always require sufficient UV/blue pre-irradiation to charge traps, and then stored carriers in the traps can be stimulated for ML under external stress. Therefore, it always requires to strictly avoid the influences of the indoor fluorescent or sunlight exposure. In this work, two strategies were adopted to rule out the potential influence of the indoor fluorescent

or sunlight exposure on the ML properties of the samples.

First, all experiments in this work including phosphor synthesis, elastomer preparation and ML measurement were carried out in a dark room illuminated by a red LED lamp covering 680-690 nm (Boxing Corp: YS-GS-MR-00, 10 W).

Second, our experiments (Figure R6) demonstrate that ML properties of the new $\text{CaF}_2:\text{Tb}^{3+}/\text{PDMS}$ elastomers are completely unaffected even when directly exposed to ultraviolet or blue light. Unlike the previously reported long-persistent luminescence phosphors, where ML always depends on the carrier detrapping in traps, ML of this new $\text{CaF}_2:\text{Tb}^{3+}/\text{PDMS}$ elastomer is excited by the contact electrification, so its ML is only related to the interaction between the phosphors and PDMS, and is completely independent of whether it has been irradiated by fluorescent or sunlight exposure containing ultraviolet or blue light.

As evidence, we conducted a serial of comparative experiments and prepared five identical $\text{CaF}_2:\text{Tb}^{3+}/\text{PDMS}$ elastomers. One was not exposed, while the others were sufficiently exposed to different lights (254/365/405 nm) and natural sunlight for 10 minutes, respectively. Accordingly, Figure R6(a) presents the ML intensities of these elastomers at the first stretch, and it can be seen that there is no significant difference. Furthermore, Figure R6(b) presents the normalized ML intensities of these $\text{CaF}_2:\text{Tb}^{3+}/\text{PDMS}$ elastomers under continuous stretching. It demonstrates that all these elastomer samples show the similar ML decrease curves. Figure R6(c) also exhibits the photographs of these elastomers under different irradiations and their ML photographs at the first stretch. It can be seen that the sufficient irradiation exposure does not significantly affect the ML properties of the $\text{CaF}_2:\text{Tb}^{3+}/\text{PDMS}$ elastomer samples. In fact, the contact-separation-induced ML of this new $\text{CaF}_2:\text{Tb}^{3+}/\text{PDMS}$ elastomer is only affected by the interaction between the phosphors and PDMS, and is not affected by the irradiation exposure. According to your significant comment and constructive suggestion, we have carefully revised our manuscript and added some appropriate discussions on the influence of these irradiation sources in the revision and Supplementary Note 8.

Figure R6. a) ML intensities of the $\text{CaF}_2:\text{Tb}^{3+}/\text{PDMS}$ elastomers at the first stretch; b) normalized ML intensities of the elastomers under continuous stretching; c) photographs of the elastomers under different irradiations and their ML photographs at the first stretch.

- The authors contend that $\text{CaF}_2:\text{Tb}^{3+}$ is more electronegative, leading to electron attraction from PDMS. However, according to the triboelectric series, PDMS is already relatively electronegative. Strategically, it might be more effective in terms of increasing contact electrification to make the ML particles more electropositive. Please provide the authors' perspective on this matter.

Response: dear reviewer, thank you very much for your significant comments. This is a very good comment that is not easy to answer. Generally, each and every material exhibits triboelectrification. To standardize its quantification, the triboelectric series for a wide range of polymers have been quantified by Prof. Z.L. Wang's group. [Ref 1] According to the triboelectric series, the PDMS is relatively electronegative compared to most polymers, but it may be not the case for the phosphor-PDMS couple. Figure R7 presents the surface electron distribution of the new $\text{CaF}_2:\text{Tb}^{3+}/\text{PDMS}$ elastomer, and it shows that the sprayed electrons are mainly concentrated on the phosphor particles. This result experimentally demonstrates

that the phosphor particles attract electrons more easily than the PDMS polymer.

Figure R7. Surface electron distribution of the CaF₂:Tb³⁺/PDMS elastomer.

At this stage, we can only give a possible explanation for the relatively electronegativity of the phosphors. Generally, the ability of a substance to attract electrons is essentially related to the elemental composition and structural characteristics of the substance itself. [Ref 2] For PDMS, it is relatively electronegative in the polymers. However, its main constituent is carbon with relatively low electronegativity. Additionally, PDMS does not contain electron-withdrawing groups such as benzene rings (C₆H₆), halogen atoms (e.g., fluorine, chlorine, bromine), or nitro groups (-NO₂) that strongly attract electrons. [Ref 3] On the contrary, inorganic phosphors usually contain non-metallic elements with higher electronegativities, such as oxygen, nitrogen, and fluorine, which strongly attract surrounding electrons [Ref 4]. Furthermore, metallic elements in inorganic materials often have low electronegativities but can lose electrons to form metal cations, which possess strong electron-attracting capabilities. In my opinion, the inorganic CaF₂:Tb³⁺ phosphors containing highly electronegative fluorine and Ca²⁺ cations should be more electronegative than the organic PDMS polymers.

According to your significant comments and constructive suggestion, we have carefully revised our manuscript and provided a possible explanation for the relatively electronegativity of the phosphors in the revision and Supplementary

Note 11. Thank you very much for your kind help again.

The references cited in the above discussions are listed as follows:

1. Zou, H., Zhang, Y., Guo, L., *et al.* Quantifying the triboelectric series. *Nat. Commun.* **10**, 1427 (2019).
 2. Briegleb, G. Electron Affinity of Organic Molecules. *Angew. Chem. Int. Ed. Engl.* **3**, 617 (1964).
 3. Politzer, P., Murray, J. S. & Clark, T. Halogen bonding: an electrostatically-driven highly directional noncovalent interaction. *Phys. Chem. Chem. Phys.* **12**, 7748 (2010).
 4. Pritchard, H. O. & Skinner, H. A. The Concept Of Electronegativity. *Chem. Rev.* **55**, 745 (1955).
6. In the explanation of the ML process, it is mentioned that after critical stretching, electrons transfer to $\text{CaF}_2:\text{Tb}^{3+}$ and subsequently return to PDMS, causing direct excitation. What is the mechanism behind the return of electrons to PDMS? Is it due to breakdown or neutralization? Since the returned electrons contribute to ML, it is important to provide a detailed explanation of this phenomenon.

Response: dear reviewer, thank you very much for your constructive comments. The physical mechanism behind the return of electrons to PDMS is very significant to understand the ML of the new $\text{CaF}_2:\text{Tb}^{3+}/\text{PDMS}$ elastomer. We sincerely apologize for not providing a more detailed explanation of this significant phenomenon. According to your constructive suggestion, we have carefully revised our manuscript and added a more detailed description of the physical process to this significant phenomenon in the revision.

The formation and recombination mechanism of the electron-hole pairs for the $\text{CaF}_2:\text{Tb}^{3+}/\text{PDMS}$ elastomer can be depicted in Figure R8. Before being compounded, the potential wells of the phosphors and PDMS are separated (i), and no electrostatic charges are created on their surfaces. Then, when the phosphors and PDMS are in close contact after being compounded, the two single potential wells become an asymmetric double potential well, and the energy barrier between the two wells is lowered (ii). Therefore, the electrons at higher levels transfer from PDMS to phosphor to maintain energy level balance. When the elastomer is

stretched to reach a critical stretching distance, the previously contacted surfaces of the phosphors and PDMS is slightly separated (iii). The electrons still remain on the surface of the phosphors, resulting in negative electrostatic charges (electrons) on the phosphors and positive electrostatic charges (holes) on the PDMS due to the contact electrification effect. At the same time, the electron-hole pairs generate a strong electrostatic field. Because the gap between the phosphors and the PDMS is very tiny, the distance between the electron and the hole is actually very short. Therefore, the negative electrons on the surface of the phosphors can be attracted back to the PDMS by the opposite charge (positive hole) in a short time. Consequently, the electron-hole recombination occurs at the phosphor-PDMS interface to release excitation energy, thereby exciting the nearby Tb^{3+} emitters for ML. This phenomenon is very similar to the electrical sparks we see when we take off a sweater in cold dry winter due to the recombination of electrostatic charges.

Figure R8. Formation and recombination mechanism of the electron-hole pairs for the $CaF_2:Tb^{3+}/PDMS$ elastomer.

7. The term "self-recovery" was originally introduced by Chandra (ref. 60) to explain the first demonstration of 100,000 MLs in the $ZnS:Cu/PDMS$ system, even without

UV/blue light pretreatment (Appl. Phys. Lett. 102, 051110 (2013)). If the authors' results emphasize the significance of this aspect, it would be beneficial to include a brief history of self-recoverable ML in the introduction to aid readers' comprehension.

Response: dear reviewer, this is a very significant comment and thank you very much for giving us such a constructive suggestion. According to your suggestion, we have carefully revised the introduction section of our manuscript and added a brief history of the self-recoverable ML to help readers understand in the revision.

In the past few decades, many ML materials have been reported, and the most well-known of which are asymmetric piezoelectric $\text{SrAl}_2\text{O}_4:\text{Eu}^{2+}$ and $\text{ZnS}:\text{Cu}/\text{Mn}^{2+}$. [Ref 1–4] However, $\text{SrAl}_2\text{O}_4:\text{Eu}^{2+}$ always requires preirradiation to charge traps before emitting ML, which severely limits its application. [Ref 5–8] ML of $\text{ZnS}:\text{Cu}/\text{Mn}^{2+}$ does not require preirradiation. [Ref 9] Particularly, Jeong and Choi reported the first demonstration of 100,000 MLs in $\text{ZnS}:\text{Cu}/\text{PDMS}$ films without preirradiation. [Ref 10] In 2013, Chandra first reported that the "self-recovery" ML of the sulfides take place by trapping of drifting carriers in piezoelectric field. [Ref 11] Consequently, ML displays seemed to be becoming a reality, and the piezoelectric sulfides are considered the best ML materials in a long period. However, the sulfides are unstable and lack multicolor luminescence, which affect their practical application. Therefore, we cannot stop exploring new high-efficiency ML materials beyond sulfides. In particular, most previous high-efficiency ML materials belong to asymmetric piezoelectric structures, and there are few studies on the materials with centrosymmetric structures. [Ref 12]

The references cited in the above introduction are listed as follows:

1. Sharma, R. & Sharma, U. Influence of fluxing agent on the luminescence properties of $\text{SrAl}_2\text{O}_4:\text{Eu}^{2+}$ nanophosphors. *J. Alloys Compd.* **649**, 440–446 (2015).
2. Tu, D., Xu, C.-N., Saito, R., Liu, L. & Yoshida, A. Influence of H_3BO_3 addition on mechanoluminescence property of $\text{SrAl}_2\text{O}_4:\text{Eu}^{2+}$. *J. Ceram. Soc. Jpn.* **125**, 648–651 (2017).
3. Zhou, T. et al. Unrevealing Temporal Mechanoluminescence Behaviors at High Frequency via Piezoelectric Actuation. *Small* **19**, 2207089 (2023).
4. Chandra, B. P., Chandra, V. K., Jha, P., Pateria, D. & Baghel, R. N. Is the

- fracto-mechanoluminescence of ZnS: Mn phosphor dominated by charged dislocation mechanism or piezoelectrification mechanism? *Luminescence* 31, 67–75 (2016).
5. Terasaki, N. & Xu, C.-N. Mechanoluminescence recording device integrated with photosensitive material and europium-doped SrAl₂O₄. *Jpn. J. Appl. Phys.* 48, 04C150 (2009).
 6. Carpenter, M. A. et al. Elastic anomalies due to structural phase transitions in mechanoluminescent SrAl₂O₄: Eu. *J. Appl. Phys* 107, (2010).
 7. Fujio, Y., Xu, C.-N. & Terasaki, N. Flexible mechanoluminescent SrAl₂O₄: Eu film with tracking performance of CFRP fracture phenomena. *Sensors* 22, 5476 (2022).
 8. Bisen, D. P. & Sharma, R. Mechanoluminescence properties of SrAl₂O₄: Eu²⁺ phosphor by combustion synthesis. *Luminescence* 31, 394–400 (2016).
 9. Wang, N. et al. Control of triboelectricity by mechanoluminescence in ZnS/Mn-containing polymer films. *Nano Energy* 90, 106646 (2021).
 10. Jeong, M. S., Song, S., Lee, S. K. & Choi, B. Mechanically driven light-generator with high durability. *Appl. Phys. Lett.* 102, (2013).
 11. Chandra, V. K., Chandra, B. P. & Jha, P. Self-recovery of mechanoluminescence in ZnS: Cu and ZnS: Mn phosphors by trapping of drifting charge carriers. *Appl. Phys. Lett.* 103, (2013).
 12. Tiwari, G. et al. Ca₂Al₂SiO₇:Ce³⁺ phosphors for mechanoluminescence dosimetry. *Luminescence* 31, 1479–1487 (2016).

Reviewer #3 (Remarks to the Author):

This manuscript reports the self-recoverable ML elastomers based on centrosymmetric fluoride phosphors and PDMS. Contact electrification is deemed to be responsible for the ML mechanism, where a contact-separation model is proposed to explain the detailed phenomena. However, the similar contact electrification induced ML in several recently-developed phosphors has been reported by the same group previously (Nano Energy 94 (2022) 106920, et.al.). Although the authors have argued the major difference (i.e., the ML resuming ratio after each hard stretching) between the present ML phenomenon and the previous ones, it is hard to convince one that this difference is so significant that deserves to be published in Nature Communications. In addition, some issues need to be considered by the authors as follows.

1. The authors keep emphasizing the ML comes from the centrosymmetric structure. Why is this so important? In my opinion, the contact electrification induced ML may happen for phosphors with different structures, as long as the contact electrification occurs.

Response: dear reviewer, thank you very much for your comments. According to your suggestion, we have added some introduction about the reason why we focused on the centrosymmetric phosphors in the revision and Supplementary Note 1.

In general, the piezoelectricity manifests only in phosphors with asymmetrical piezoelectric structures, whereas the contact electrification occurs in almost all phosphors regardless of structural symmetry. However, because piezoelectricity in piezoelectric structures is significantly stronger than contact electrification, it is very difficult to observe the contact electrification effect in piezoelectric materials. Consequently, almost all previously reported MLs in piezoelectric materials were attributed to the piezoelectricity rather than contact electrification, as exhibited in Table R3. Even the few centrosymmetric phosphors without piezoelectric effect emit ML under stress, and their MLs are still attributed to the local piezoelectricity due to doping (marked “local pieze.” in Table R3). In general, it is easier to observe the contact electrification effect at the interface of two insulators. The most well-known example of contact electrification appearing in textbooks is the rubbing of an animal’s fur against a plastic rod. Since both the plastic rod and animal’s fur are

insulators, we can observe some typical static electricity phenomena due to contact electrification, such as static sparks and beeping noise. Correspondingly, centrosymmetric phosphors with large dielectric constant usually exhibit physical properties similar to insulators, and therefore we can observe stronger contact electrification effect in the centrosymmetric phosphors. For example, our group has reported a series of ML elastomers based on PDMS and centrosymmetric phosphors such as $\text{Sr}_3\text{Al}_2\text{O}_6:\text{Eu}^{3+}$, $\text{Y}_3\text{Al}_5\text{O}_{12}:\text{Ce}^{3+}$, $\text{Lu}_3\text{Al}_5\text{O}_{12}:\text{Ce}^{3+}$ and $\text{Gd}_5\text{Ga}_3\text{O}_{12}:\text{RE}^{3+}$ [Ref 1-4], which show much stronger ML than $\text{ZnS}:\text{Cu},\text{Mn}^{2+}/\text{PDMS}$ elastomer. Accordingly, it is worthwhile to research the ML properties of the centrosymmetric phosphors. However, current research on ML is mostly focused on the asymmetric piezoelectric materials. We can't always follow in the footsteps of our predecessors. Our world requires different perspectives and different voices. Centrosymmetric phosphors show more excellent ML performance and better development potential, so our group decides to focus on the ML phosphors with centrosymmetric structures, which is significantly different from most other researchers in the field of ML, and is precisely the significant value of our existence. Thank you very much for giving us a significant chance to show a different voice.

The references cited in the above introduction are listed as follows:

1. Wu, C. et al. Efficient Mechanoluminescent Elastomers for Dual-Responsive Anticounterfeiting Device and Stretching/Strain Sensor with Multimode Sensibility. *Adv. Funct. Mater.* 28, 1803168 (2018).
2. Ma, Z. et al. Mechanics-induced triple-mode anticounterfeiting and moving tactile sensing by simultaneously utilizing instantaneous and persistent mechanoluminescence. *Mater. Horiz.* 6, 2003–2008 (2019).
3. Zhou, J. et al. An ultra-strong non-pre-irradiation and self-recoverable mechanoluminescent elastomer. *Chem. Eng. J.* 390, 124473 (2020).
4. Wang, J. et al. Contact Electrification Induced Multicolor Self-Recoverable Mechanoluminescent Elastomer for Wearable Smart Light-Emitting Devices. *Adv. Opt. Mater.* 11, 2203112 (2023).

Table R3. Crystal structure, piezoelectricity (piezoe.) -induced ML, contact-electrification-induced ML for previously reported ML materials.

Crystal system	Point group	symmetry order	Piezoe. structure	Material hosts	Piezoe.- induced ML	Contact-electrification-induced ML
Triclinic	1	1	√	CaAl ₂ Si ₂ O ₈ SrAl ₂ Si ₂ O ₈ SrSi ₂ O ₂ N ₂	√	
	$\bar{1}$	2	×		×	
	m	2	√		√	
Monoclinic	2	2	√	SrAl ₂ O ₄ SrMg ₂ (PO ₄) ₂ Ca ₂ Nb ₂ O ₇ Sr ₂ Nb ₂ O ₇	√	
	2/m	4	×		×	
	mm2	4	√	Ca ₃ Ti ₂ O ₇ BaSi ₂ O ₂ N ₂ SrZn ₂ S ₂ O	√	
Orthorhombic	222	4	√	Sr ₃ Sn ₂ O ₇ CaZr(PO ₄) ₂	√	
	mmm	8	×	NaNbO ₃ BaZnOS CaNb ₂ O ₆	√ (local piezoe.)	
	3	3	√	Zn ₂ (Ge,Si)O ₄	√	
Trigonal	$\bar{3}$	6	×		×	
	3m	6	√	LiNbO ₃	√	
	32	6	√		√	
	$\bar{3}m$	12	×		×	
Tetragonal	4	4	√		√	
	$\bar{4}$	4	√		√	
	4/m	8	×		×	
	4mm	8	√	(Ba,Ca)TiO ₃	√	
	$\bar{4}2m$	8	√	Ca ₂ Al ₂ SiO ₇ CaYAl ₃ O ₇ Ca ₂ MgSi ₂ O ₇ Sr ₂ MgSi ₂ O ₇	√	
	422	8	√		√	

	$4/mmm$	16	×	$\text{Ca}_3\text{Nb}_2\text{O}_8$	√ (local piezoe.)
	6	6	√		√
	$\bar{6}$	6	√		√
	6/m	12	×		×
Hexagonal	$\bar{6}m2$	12	√		√
	6mm	12	√	ZnS	√
	622	12	√		√
	6/mmm	24	×		×
	23	12	√		√
	$m\bar{3}$	24	×		×
	$\bar{4}3m$	24	√		√
	432	24	×		×
Cubic				ZnAl_2O_4	√ (tribe.)
				MgGa_2O_4	√ (tribe.)
	$m\bar{3}m$	48	×	ZnGa_2O_4	√ (tribe.)
				$\text{Y}_3\text{Al}_5\text{O}_8$	√
				$\text{Lu}_3\text{Al}_5\text{O}_8$	
				$\text{Gd}_5\text{Ga}_3\text{O}_{12}$	√

2. According to the authors, the separation between the phosphor and PDMS is important for the ML, since the contact-electrification-induced electrons are transferred to the phosphor when the CaF_2 -PDMS interface is separated. If this is true, how to explain the ML from scratching? I don't see that there would be any separation between CaF_2 and PDMS when scratching happens.

Response: dear reviewer, thank you very much for your comments. Generally, we see nothing doesn't mean nothing happened. In this case, although it seems that we cannot "see" the scratching-induced separation between the $\text{CaF}_2:\text{Tb}^{3+}$ phosphors and PDMS with our naked eyes, it does occur! However, we are very sorry that we did not provide a detailed explanation for this significant issue. According to your significant comments, we have carefully revised our manuscript and added some appropriate explanation for this issue in the revision and Supplementary Note 13.

In this case, the entire elastomer is stressed under stretching, while only a very

tiny area in contact with the nail is stressed under scratching. Therefore, the stress-bearing area of the elastomer under stretching is much larger than that under scratching. Consequently, even if the same pressure (P) is applied, the stress ($F = P \times S$) under stretching must be much greater than the stress under scratching due to the much larger stress-bearing area (S). Accordingly, Table R4 shows the stretching stresses and scratching stresses to the $\text{CaF}_2:\text{Tb}^{3+}/\text{PDMS}$ elastomer under different pressures. For example, it shows that when the pressure applied to the elastomer is 0.2341 MPa, the stretching stress (2.5571 N) is even more than 20 times the scratching stress (0.1176 N). Consequently, we always feel that the stress under scratching is very “slight” and the stress under stretching is very “hard”, but in fact the pressure on the elastomer is still the same for both models. In this case, we may think we don't “see” the separation because we feel the stress under scratching is very “slight”, but in fact this is an illusion. As mentioned in the above example, although the scratching stress applied to the $\text{CaF}_2:\text{Tb}^{3+}/\text{PDMS}$ elastomer is as small as 0.117592 N, the local pressure on the elastomer is still as high as 0.23406 MPa, which is sufficient to induce strong strain and tiny interface separation between the phosphor particles and PDMS.

As evidences, Figure R9 shows the microscope image of the $\text{CaF}_2:\text{Tb}^{3+}/\text{PDMS}$ elastomer after scratching 200 times. It shows that the edge area of the phosphor particles in the elastomer is clearly dyed red corresponding to the tiny gaps (separation) at the interface between the phosphors and PDMS. Furthermore, Figure R10 depicts that the scratching-induced strain in the elastomer and the tiny gaps (separation) between the phosphor particles and PDMS under scratching based on finite element calculation. It presents that during the scratch, the scratching-induced strain (green region) of the elastomer can be clearly observed, resulting in the nearby separation (white region) between the phosphor particles and PDMS. However, the separation under scratching is clearly slighter, and therefore the scratching-induced damages are also slighter, corresponding to a weaker ML and a more stable ML under continuous scratching.

Table R4. Stretching stresses and scratching stresses to the $\text{CaF}_2:\text{Tb}^{3+}/\text{PDMS}$ elastomer under different pressures.

Pressure (MPa)	Stretching stress (N)	Scratching stress (N)
0.1261	1.3778	0.0634
0.2341	2.5571	0.1176
0.3304	3.6101	0.1660
0.4215	4.6051	0.2118
0.5074	5.5434	0.2549
0.5926	6.4743	0.2977
0.6798	7.4264	0.3415
0.7729	8.4442	0.3883
0.8799	9.6127	0.4421
1.0013	10.9390	0.5030

Figure R9. Microscope image of the $\text{CaF}_2:\text{Tb}^{3+}/\text{PDMS}$ elastomer after scratching 200 times.

Figure R10. Strain of the elastomer and the separation between the phosphors and PDMS under scratching based on finite element calculation.

3. What technique has been adopted to acquire the surface morphology and surface electron distribution in Fig.3C?

Response: dear reviewer, thank you very much for your comments. We sincerely apologize for not introducing the technique adopted to acquire the surface morphology and surface electron distribution, as shown in Figure 3c. According to your significant comments, we have carefully revised the experiment section of our manuscript. In particular, we have provided a brief introduction to the AFM and EFM techniques in the revision and Supplementary Note 16.

Generally, the surface morphology is measured by an atomic force microscope (AFM) and the surface electron distribution can be measured by an electrostatic force microscope (EFM). In this work, both the surface morphology and surface electron distribution of the samples were obtained by using an Oxford Cypher S AFM Microscope. AFM is a broader technique that encompasses various force interactions, while EFM is a specialized mode of AFM that specifically investigates electrostatic forces. The principle of AFM is based on the detection of forces, primarily the van der Waals forces, between the atoms on the surface of the sample and the atoms on the probe tip. These forces cause the deflection of the cantilever,

which is monitored using a laser beam deflection or an array of piezoresistive sensors. By precisely controlling the tip-sample distance, the topography of the surface can be mapped with high resolution. EFM is a specialized mode of AFM that focuses on the measurement of electrostatic forces between the probe tip and the sample surface. It provides information about the local electrical properties and surface charge distribution of the sample.

4. The authors argue that electrons transfer from PDMS to CaF₂ as long as the CaF₂ and PDMS are compounded. To this point, is the friction not necessary for the triboelectrification?

Response: dear reviewer, thank you very much for your significant comments on contact electrification i.e., triboelectrification. It may be difficult for us to think intuitively, however it is a widely accepted fact that contact electrification, i.e., triboelectrification, does not necessarily require friction, and it is not a viewpoint proposed by us. It is well known that the concept of the contact electrification was proposed by Prof. Zhonglin Wang and has been introduced in detail in many important references. [Ref 1-4] At present, it has been widely accepted that the charges are produced due to physical contact, and mechanical friction is not necessary although it can aid in delivering the charges. (The original text of this statement appears on page 35 of the reference 3 listed below.) However, we are very sorry that we did not provide more details on this significant issue. To help broader readers better understand the contact-separation-induced self-recoverable ML mechanism of this new CaF₂:Tb³⁺/PDMS elastomer, it is necessary to make an explicit introduction to the contact-separation process based on contact electrification. Accordingly, we have carefully revised our manuscript and added a brief introduction to the role of friction on the triboelectrification i.e., contact electrification in the revision and Supplementary Note 12 based on your comments.

Figure R11. Formation and recombination mechanism of the electron-hole pairs for the $\text{CaF}_2:\text{Tb}^{3+}/\text{PDMS}$ elastomer.

Generally, contact electrification is the scientific term for triboelectrification. The formation and recombination mechanism of the electron-hole pairs for the $\text{CaF}_2:\text{Tb}^{3+}/\text{PDMS}$ elastomer can be depicted in Figure R11. First (i), before being compounded, the potential wells of the phosphors and PDMS are separated, and no electrostatic charges are created on their surfaces. However, owing to the different energy bands of each material, the energy of the occupied surface states of the PDMS is higher than that of the unoccupied surface states of the phosphors. Second (ii), when the phosphors and PDMS are in close contact after being compounded, their atomic electron clouds overlap to form covalent bonds. The two single potential wells become an asymmetric double potential well, and the energy barrier between the two wells is lowered due to the strong electron-cloud overlap. Consequently, the electrons at higher levels transfer from PDMS to phosphors to maintain energy level balance even if there is no friction between the phosphor and PDMS. At this stage, no electrostatic charges are created due to the formation of new energy level balance of the asymmetric double potential well. Third (iii), when the elastomer is stretched, the previously contacted surfaces of the phosphors and PDMS is slightly separated. However, the electrons still remain on the surface of

phosphors, resulting in negative electrostatic charges (electrons) on the phosphors and positive electrostatic charges (holes) on the PDMS due to contact electrification effect. [Ref 1-4] At the same time, the electron-hole pairs generate a strong electrostatic field. Because the gaps between the phosphors and the PDMS are very tiny, the distance between the electron and hole is actually very short as well. Therefore, the negative electrons on the surface of the phosphors can be attracted back to the PDMS by the opposite charge (positive hole) in a short time. Consequently, the electron-hole recombination occurs at the phosphor-PDMS interfaces to release sufficient excitation energy, thereby exciting the nearby Tb^{3+} emitters for ML. In our lives, this electron-hole recombination generally occurs to induce some interesting static electricity phenomena, such as static sparks and beeping noise. For example, we may see the bright static sparks when we take off a sweater on a dry winter night. According to the above results and discussions, the contact-separation-induced ML mechanism of the new $CaF_2:Tb^{3+}/PDMS$ elastomer can be depicted in Figure R12.

Figure R12. Contact-separation-induced ML mechanism of the new $CaF_2:Tb^{3+}/PDMS$ elastomer.

The references cited in the above discussions are listed as follows:

1. Wang, S., Lin, L. & Wang, Z. L. Nanoscale Triboelectric-Effect-Enabled Energy Conversion for Sustainably Powering Portable Electronics. *Nano Lett.* **12**, 6339–6346 (2012).
2. Bae, J. *et al.* Flutter-driven triboelectrification for harvesting wind energy. *Nat. Commun.* **5**, 4929 (2014).
3. Wang, Z. L. & Wang, A. C. On the origin of contact-electrification. *Materials Today* **30**, 34–51 (2019).
4. Liu, Z. *et al.* Fabrication of triboelectric polymer films via repeated rheological forging for ultrahigh surface charge density. *Nat. Commun.* **13**, 4083 (2022).
5. According to the authors, the rapid electron-hole recombination happens only after the separation between CaF₂ and PDMS. If the electrons transfer from the PDMS to CaF₂, holes should be left in PDMS. In this regard, I cannot understand how the electron-hole recombination may happen when electrons and holes are spatially separated.

Response: dear reviewer, thank you very much for pointing out this issue. We sincerely apologize for not providing a more detailed description on this issue. Similar to the previous comment, this comment still involves the physical mechanism of contact electrification. Therefore, in order to provide a more complete and clear response to your question, we have merged the response of this comment with the previous comment. We believe it will help the readers understand the mechanism of the electron-hole recombination. According to your suggestion, we have carefully revised our manuscript and have added some discussion on the electron-hole recombination in the revision.

The physical contact-separation processes of the CaF₂:Tb³⁺/PDMS elastomer before and after contact have been introduced in above Figure R12(i) and Figure R12(ii). In response to this comment, we focus here on the physical mechanism after separation of the phosphors and PDMS. Accordingly, the previous Figure R12 is also exhibited in this response as the Figure R13 below. It shows in Figure R13(iii) that when the elastomer is stretched, the previously contacted surfaces of the

phosphors and PDMS is slightly separated. However, the electrons still remain on the surface of phosphors, resulting in negative electrostatic charges (electrons) on the phosphors and positive electrostatic charges (holes) on the PDMS due to contact electrification effect. [Ref 1-4] At the same time, the electron-hole pairs generate a strong electrostatic field. Because the gaps between the phosphors and the PDMS are very tiny, the distance between the electron and hole is actually very short as well. Therefore, the negative electrons on the surface of the phosphors can be attracted back to the PDMS by the opposite charge (positive hole) in a short time. Consequently, the electron-hole recombination occurs at the phosphor-PDMS interfaces to release sufficient excitation energy, thereby exciting the nearby Tb^{3+} emitters for ML. In our lives, this electron-hole recombination generally occurs to induce some interesting static electricity phenomena, such as static sparks and beeping noise. For a typical example, we may see the bright static sparks when we take off a sweater on a dry winter night.

Figure R13. Formation and recombination mechanism of the electron-hole pairs for the $CaF_2:Tb^{3+}/PDMS$ elastomer.

The references cited in the above discussions are listed as follows:

1. Wang, S., Lin, L. & Wang, Z. L. Nanoscale Triboelectric-Effect-Enabled Energy Conversion for Sustainably Powering Portable Electronics. *Nano Lett.* **12**, 6339–6346 (2012).
2. Bae, J. *et al.* Flutter-driven triboelectrification for harvesting wind energy. *Nat. Commun.* **5**, 4929 (2014).
3. Wang, Z. L. & Wang, A. C. On the origin of contact-electrification. *Materials Today* **30**, 34–51 (2019).
4. Liu, Z. *et al.* Fabrication of triboelectric polymer films via repeated rheological forging for ultrahigh surface charge density. *Nat. Commun.* **13**, 4083 (2022).

REVIEWER COMMENTS

Reviewer #1 (Remarks to the Author):

The authors have addressed the reviewers' concerns very well and the work can be accepted for publication.

Reviewer #2 (Remarks to the Author):

The authors have fully revised the paper according to my comments. Therefore, I recommend its publication in Nature Communications.

Reviewer #3 (Remarks to the Author):

The authors have tried their best to address all the problems. However, I still have some concerns about the proposed mechanism.

1. ML happens during the stretching process. Is there any ML signal may be detected when the stretched compound is released?

2. According to the authors, when the elastomer is stretched to reach a critical stretching distance, the previously contacted surfaces between the phosphors and PDMS can be slightly separated, which leads to the happening of ML. If one holds the elastomer in the stretched position, part of the phosphors may be suspended in the air (no contact between this part of phosphors and PDMS), or new interface may come into being between this part of phosphors and different PDMS (so the contact electrification may occur immediately at this stage)? To this point, what can we expect to happen given these two scenarios?

RESPONSE TO REVIEWERS' COMMENTS

Reviewer #3 (Remarks to the Author):

The authors have tried their best to address all the problems. However, I still have some concerns about the proposed mechanism.

1. ML happens during the stretching process. Is there any ML signal may be detected when the stretched compound is released?

Response: dear reviewer, thank you very much for your significant comments and we sincerely apologize for not providing a detailed description of this issue. Accordingly, Figure R1(i) exhibits the ML intensities and stretching distances of the $\text{CaF}_2:\text{Tb}^{3+}/\text{PDMS}$ elastomer during a single stretching. It indicates that the ML of the $\text{CaF}_2:\text{Tb}^{3+}/\text{PDMS}$ elastomer only happens once during the stretching process, and we can never observe any ML when the stretched elastomer is released. However, as a comparison, the ML of the $\text{ZnS}:\text{Cu}/\text{PDMS}$ elastomer can be detected during both the stretching and releasing processes, as shown in Figure R1(ii). Since the ML of $\text{CaF}_2:\text{Tb}^{3+}/\text{PDMS}$ elastomer is due to contact electrification, it is understandable that its ML can be only observe once when the contacted surfaces of the phosphors and PDMS are separated during the stretching process. However, the ML of the $\text{ZnS}:\text{Cu}/\text{PDMS}$ elastomer is essentially associated with strain-induced piezoelectricity. Generally, strain can be created during both stretching and releasing processes of the elastomers, resulting in piezoelectricity. Therefore, the ML of the $\text{ZnS}:\text{Cu}/\text{PDMS}$ elastomer can be also observed during both the stretching and releasing processes. According to your significant comments and constructive suggestion, we have carefully revised our manuscript and added a more intuitive description of this issue in the revision and Supplementary Note 11 in the supporting information.

Figure R1. ML intensities and stretching distances of the $\text{CaF}_2:\text{Tb}^{3+}/\text{PDMS}$ (i) and $\text{ZnS}:\text{Cu}/\text{PDMS}$ (ii) elastomers during a single stretching.

2. According to the authors, when the elastomer is stretched to reach a critical stretching distance, the previously contacted surfaces between the phosphors and PDMS can be slightly separated, which leads to the happening of ML. If one holds the elastomer in the stretched position, part of the phosphors may be suspended in the air (no contact between this part of phosphors and PDMS), or new interface may come into being between this part of phosphors and different PDMS (so the contact electrification may occur immediately at this stage)? To this point, what can we expect to happen given these two scenarios?

Response: dear reviewer, thank you very much for your significant comments and we are very sorry that we did not provide a detailed description of this issue. As shown in Figure R2, the areas marked with red dotted lines correspond to the ML performances of the $\text{CaF}_2:\text{Tb}^{3+}/\text{PDMS}$ (i) and $\text{ZnS}:\text{Cu}/\text{PDMS}$ (ii) elastomers when one holds the elastomer in the stretched position. The experiment results show that

we cannot observe any ML when the $\text{CaF}_2:\text{Tb}^{3+}/\text{PDMS}$ (i) elastomer is held in the stretched position, while the ML can be still detected when the $\text{ZnS}:\text{Cu}/\text{PDMS}$ (ii) elastomers is held. According to contact electrification, carrier recombination occurs only at the moment of surface separation during the stretching process. When the elastomer is held in the stretched position, the surfaces of the phosphors and PDMS have been separated during the previous stretching process and therefore we can no longer observe the ML. However, it is well known that piezoelectricity is due to the stress-induced strain in a material. Even if the elastomer is held in the stretched position, there is still stress-induced strain in the elastomer, so we can still observe the piezoelectricity-induced ML of the $\text{ZnS}:\text{Cu}/\text{PDMS}$ (ii) elastomer.

Figure R2. ML intensities and stretching distances of the $\text{CaF}_2:\text{Tb}^{3+}/\text{PDMS}$ (i) and $\text{ZnS}:\text{Cu}/\text{PDMS}$ (ii) elastomers during a single stretching.

Figure R3. Schematic diagram and finite element stress (von Mises) simulations of the $\text{CaF}_2:\text{Tb}^{3+}/\text{PDMS}$ elastomer under a two-dimensional stretching

Moreover, Figure R3 presents the schematic diagram and finite element stress (von Mises) simulations of the $\text{CaF}_2:\text{Tb}^{3+}/\text{PDMS}$ elastomer under a two-dimensional stretching. It shows that when the elastomer is stretched horizontally in the x direction (i), the internal phosphor particles and PDMS will be separated by the horizontal stretching stress, while they contact due to the contraction of the elastomer in the vertical direction (iii). Then, when the elastomer is vertically stretched in the y direction (ii), the previously contacted surfaces of the phosphor and PDMS in the vertical direction will be separated under stretching stress, while the previously separated surfaces in the horizontal direction will be recontacted due to the contraction of the elastomer, as shown in Figure R3(iv). This result shows that surface separation occurs during the stretching process, resulting in carrier recombination and ML of the elastomers, while surface contact occurs during the

contracting/releasing process, corresponding to electron transfer due to contact electrification. Therefore, nothing will happen when the $\text{CaF}_2:\text{Tb}^{3+}$ /PDMS elastomer is held in the stretched position. According to your significant comments, we have carefully revised our manuscript and added a more detailed introduction of this issue in the revision and Supplementary Note 11 in the supporting information. Thank you very much for your kind help and constructive suggestion.

REVIEWERS' COMMENTS

Reviewer #3 (Remarks to the Author):

The authors have addressed all the problems and the manuscript can be accepted for publication in the present form.